# Divergent self-association properties of paralogous proteins TRIM2 and TRIM3 regulate their E3 ligase activity

Diego Esposito [1], Jane Dudley-Fraser [1], Acely Garza-Garcia [2] & Katrin Rittinger [1] ✉

Tripartite motif (TRIM) proteins constitute a large family of RING-type E3 ligases that share a conserved domain architecture. TRIM2 and TRIM3 are paralogous class VII TRIM members that are expressed mainly in the brain and regulate different neuronal functions. Here we present a detailed structure-function analysis of TRIM2 and TRIM3, which despite high sequence identity, exhibit markedly different self-association and activity profiles. We show that the isolated RING domain of human TRIM3 is monomeric and inactive, and that this lack of activity is due to a few placental mammal-specific amino acid changes adjacent to the core RING domain that prevent self-association but not E2 recognition. We demonstrate that the activity of human TRIM3 RING can be restored by substitution with the relevant region of human TRIM2 or by hetero-dimerization with human TRIM2, establishing that subtle amino acid changes can profoundly affect TRIM protein activity. Finally, we show that TRIM2 and TRIM3 interact in a cellular context via their filamin and coiled-coil domains, respectively.

Modification of proteins with ubiquitin is a highly versatile mechanism to allow a rapid cellular response to many stimuli, such as regulation of protein homeostasis or clearance of infections[1]. Ubiquitination is mediated by the sequential activities of three enzymes, the last of which, the E3 ubiquitin ligase, provides specificity through substrate selection and determines the type of ubiquitin modification, either directly or, in the case of RING E3 ubiquitin ligases, through partnering with a linkage-type specific E2 conjugating enzyme[2,3]. The tripartite motif (TRIM) family constitutes the largest subfamily of RING-type E3s with more than 70 members in humans[4,5]. They have been associated with a wide range of biological processes including apoptosis, autophagy, cellular differentiation, DNA repair, tumor suppression, and innate immune responses[6–8]. TRIM family members are characterized by the presence of an N-terminal tripartite motif, also referred to as RBCC, consisting of a RING domain, the heart of TRIM E3 catalytic activity, followed by one or two B-box domains, and a long coiled-coil stem that is responsible for antiparallel homodimerization of TRIM

proteins[8,9]. A variable C-terminal domain with a role in substrate recruitment follows the RBCC motif and it is used to classify the TRIM family into 11 classes[10]. While many reported cellular roles of TRIM family members have been linked to their E3 ligase activity, there is an increasing number of functions that appear ubiquitination independent or may require additional protein partners or stimuli for activity. At present, the mechanistic details of TRIM protein catalytic activities have been characterized only for a limited number of family members and it is unknown which TRIM proteins are bona fide E3 ligases, and which may have alternative functions.

TRIM2, 3, 32, and 71 are members of TRIM class VII, a class defined by the presence of a C-terminal NHL domain. In all members, except TRIM32, the NHL domain is preceded by an immunoglobulin-like filamin domain; TRIM71 is the only member with two B-boxes. Some class VII TRIMs possess RNA-binding properties via their NHL domain[11]. Human members have been associated with multiple cellular roles that include cell-cycle regulation, tumor suppression, cell differentiation,

[1]Molecular Structure of Cell Signalling Laboratory, The Francis Crick Institute, 1 Midland Road, London NW1 1AT, UK. [2]Mycobacterial Metabolism and Antibiotic Research Laboratory, The Francis Crick Institute, 1 Midland Road, London NW1 1AT, UK. ✉e-mail: katrin.rittinger@crick.ac.uk

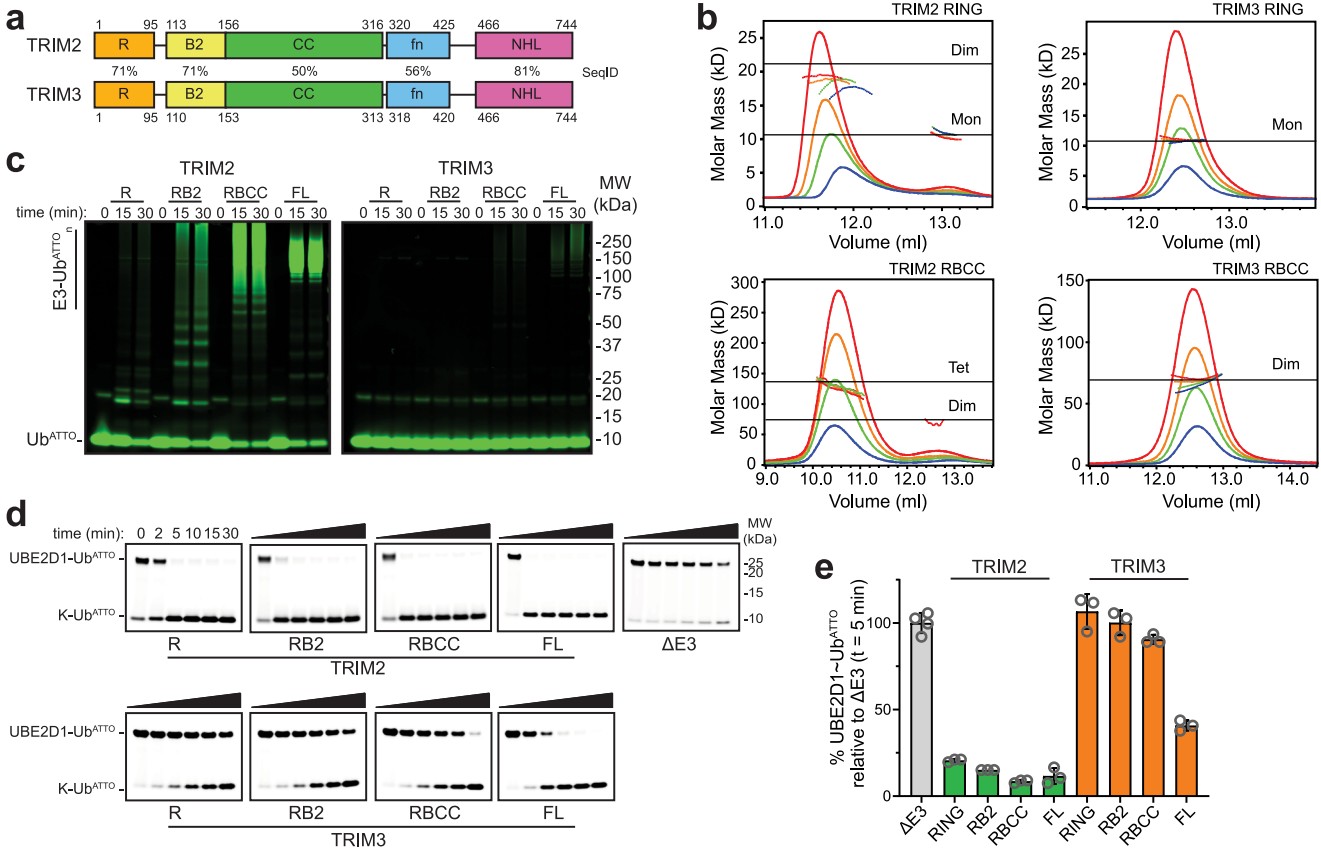

**Fig. 1 | TRIM2 and TRIM3 oligomeric state and catalytic activity. a** Domain organization of the E3 ligases. The numbering is based on the constructs used in this study for the RING (R), RING-B-box2 (RB2) and tripartite motif (RBCC), the TRIM2 filamin (fn) and TRIM3 filamin boundaries are based on the TRIM3 fn NMR structure (7O0B) and NHL boundaries on the TRIM2 (7QRV) and TRIM3 (7QRW) NHL domain crystal structures. Domains primary sequence identity (SeqID) is reported in between the cartoon representation. **b** SEC-MALLS analysis of different constructs of TRIM2 and TRIM3. The proteins were analyzed at a concentration of 5 (red), 3 (orange), 2 (green) and 1 mg/ml (blue). **c** Auto-ubiquitination and

(**d**) UBE2D1-Ub$^{ATTO}$ discharge assay with different TRIM2 and TRIM3 constructs. Reactions were carried over a 30-min time interval. The LI-COR Odyssey One-Color was used as molecular weight marker. **e** Quantification of the UBE2D1-Ub$^{ATTO}$ band after 5-min reaction time reported as normalized intensity relative to the experiment in the absence of E3. The values are presented as mean ± SD of three independent experiments ($n = 3$) for the TRIM2 and TRIM3 fragments and full length proteins and of four independent experiments ($n = 4$) in the absence of E3. Individual data are shown as gray circles. Source data is provided as a Source Data file.

and neurodevelopment[12]. The crystal structure of the NHL domains of human TRIM2 (7QRV), TRIM3 (7QRW), and TRIM71 (7QRX) show a six-bladed β-propeller, where each blade is made of four antiparallel β-sheets[13]. A similar tertiary fold is observed for the NHL domains of the fruit fly TRIM3 orthologue Brat (4ZLR)[14] and of the zebrafish TRIM71 orthologue, LIN-41 (6FQL)[15]. Both non-human NHL domain structures were determined in complex with small RNAs, whereas structural characterization of a human TRIM NHL bound to oligonucleotides has remained elusive.

Human TRIM2 and TRIM3 are highly homologous proteins: their RING and B-box domains share 71% sequence identity whilst their C-terminal NHL domains share 81% sequence identity (Fig. 1a). They are both predominantly expressed in the brain, but different functions have been attributed to the two proteins. TRIM2 is present in the cytoplasm and in cytoplasmic filaments and it has been reported to enhance axon specification during neuronal polarization[16]. It binds Myosin V via its NHL domain contributing to the modulation of neural plasticity in the central nervous system[17]. TRIM2 induces ubiquitination-dependent degradation of multiple substrates including the pro-apoptotic protein Bim, thereby conferring neuroprotection in rapid ischemic events[18], and the neurofilament light chain, with TRIM2 knockout mice showing gradual neurodegeneration and axon swelling[19]. Mutations in the TRIM2 gene have significant implications in neurodegeneration such as the progressive muscle and sensory loss observed in Charcot-Marie-Tooth disease associated with the D667A

mutation[20]. Furthermore, increased expression of TRIM2 has been observed in Alzheimer's disease as a consequence of a decrease in miRNAs able to downregulate TRIM2 mRNA[21].

TRIM3 is a component of the cytoskeleton-associated recycling or transport complex, which contains Myosin V[22], and polyubiquitinates γ-actin, regulating neuronal plasticity, learning, and memory[23]. It interacts with the neuronal kinesin KIF21B via its tripartite motif, is involved in the regulation of microtubule dynamics, synapse function and neuronal morphology and its deletion impairs protein trafficking in neurons[24]. Enhanced expression of TRIM3 has been associated with schizophrenia and it has been proposed as a novel biomarker for early diagnosis of Parkinson's disease[25]. In addition to its neuronal functions, TRIM3 mediates K63-linked poly-ubiquitination of TLR3 upon stimulation with poly(I:C)[26], and functions as a brain tumor suppressor gene via attenuation of Notch signaling and suppression of c-myc expression[27]. Moreover, TRIM3 suppresses cell growth by ubiquitination of the cell-cycle regulator p21 and is downregulated in various cancer types including liver cancer, esophageal cell carcinoma and colon cancer[28,29].

Studies of TRIM2 and TRIM3 have thus far focused on their cellular role, and little is known regarding the mechanistic basis of their observed functions and regulation of their E3 ligase activity. In all TRIM proteins studied in molecular detail, RING homodimerization is necessary for E3 ligase activity. The RING dimer stabilizes a closed conformation of the E2-Ub conjugate in which each ubiquitin molecule

contacts both RING domains, priming the thioester bond for ubiquitin transfer[30]. Accordingly, the RING domain of class VII member TRIM32 forms a constitutive dimer, which is able to auto-ubiquitinate in the presence of UBE2D1 and to synthesize unanchored K63-linked poly-ubiquitin chains with the UBE2N/UBE2V1 heterodimer[31].

In this study, we have carried out a detailed biochemical and structural characterization of TRIM2 and TRIM3 to elucidate the mechanistic principles guiding the activity of class VII TRIMs. We present the crystal structure of the dimeric RING domain of TRIM2 bound to the UBE2D1-Ub conjugate and show that despite their high sequence identity TRIM2 and TRIM3 RING and RBCC domains exhibit different self-association properties that impact their catalytic activity. Whilst TRIM2 RING E3 activity resembles that reported for TRIM32, TRIM3 RING in isolation does not possess ligase activity and does not self-associate. We demonstrate that TRIM3 catalytic activity is partially restored in the full-length protein and it is completely recovered upon enforced homodimerization of its RING domain or heterodimerization with TRIM2. Furthermore, we show that endogenous TRIM2 and TRIM3 interact in brain-derived cell lines, suggesting that these two TRIMs might influence each other's activity and function. Based on phylogenetic analysis and the data presented here, we speculate that in placental mammals the catalytic properties of TRIM2 and TRIM3 have diverged to support an evolving brain acquiring higher functions.

## Results

### TRIM2 and TRIM3 have different self-association patterns

The oligomeric state of TRIM2 and TRIM3 RING, RING-B-box2 (RB2) and RBCC constructs was assessed by analytical size-exclusion chromatography coupled with multi-angle laser light scattering (SEC-MALLS) (Fig. 1b and Supplementary Fig. 1a). We observed concentration-dependent self-association for the TRIM2 RING domain, with a strong tendency towards a dimeric form, separate from a monomeric species eluting at higher retention volumes. In contrast, across the range of concentrations explored, the TRIM3 RING domain elutes as a symmetric peak with a molecular mass corresponding to that of a monomeric species. The presence of the B-box2 does not alter the detected self-association pattern for either RING: whilst TRIM2 RB2 shows a tendency to dimerize, TRIM3 RB2 is monomeric in the range of concentrations explored highlighting that the B-box does not induce higher order self-association (Supplementary Fig. 1a). As previously observed for TRIM32[31], the tripartite motif of TRIM2 associates into a homo-tetramer whilst the TRIM3 RBCC forms a CC-mediated dimer (Fig. 1b).

### The effect of oligomerization on TRIM2 and TRIM3 catalytic activity

To investigate whether the different oligomerization patterns observed for TRIM2 and TRIM3 are reflected in their catalytic activity, we performed auto-ubiquitination assays using the RING, RB2, RBCC and full-length constructs of both, TRIM2 and TRIM3, with the promiscuous E2 conjugating enzyme UBE2D1 and fluorescently labeled Ub[ATTO] (Fig. 1c). The assays show that whilst the RB2, RBCC and full-length constructs of TRIM2 have strong auto-ubiquitination activity, for TRIM3 only the full-length protein manifests some E3 activity with faint bands from fluorescent Ub[ATTO] appearing after 30 min. Auto-ubiquitination of full length TRIM3 could not be detected on a Coomassie-stained gel (Supplementary Fig. 1b). Similarly, the isolated RING domain of TRIM2 shows much reduced apparent catalytic activity compared to that exhibited by longer constructs. Since auto-ubiquitination assay read-outs are affected by the number of available acceptor lysine residues, we used a lysine-discharge assay to monitor the ability of a given E3 construct to transfer ubiquitin to free lysine from a pre-loaded E2-Ub[ATTO] conjugate. All TRIM2 constructs, including the RING domain, discharge ubiquitin within a few minutes, with catalytic activity slightly enhanced for the longer RBCC and full-length

constructs (Fig. 1d−e and Supplementary Fig. 1c). In comparison, the RING and RB2 constructs of TRIM3 are inactive whilst the tripartite motif and full length TRIM3 discharge the conjugate, although with significantly lower activity than TRIM2 (Fig. 1d−e and Supplementary Fig. 1c).

To investigate if TRIM2 and TRIM3 might have preferential activity with other ubiquitin-conjugating enzymes, we tested each full-length protein in an "E2[scan]" (Ubiquigent, Dundee) experiment containing all E2 conjugating enzymes (Supplementary Fig. 2). Our data show that after one hour, TRIM2 is active with a number of E2s, including those in the UBE2D and UBE2E families and the UBE2N/UBE2V1 heterodimer, whilst TRIM3 shows weak E3 ligase activity only with UBE2D family members.

### Structure of the active TRIM2 RING/ UBE2D1-Ub complex

To gain molecular insight into the role of self-association in the observed difference in activity between the highly homologous RING domains of TRIM2 and TRIM3, we solved the crystal structure of TRIM2 RING in complex with UBE2D1 conjugated to ubiquitin (UBE2D1-Ub). Crystals of the complex, refined to 2.5 Å, belong to the space group $P1\,21\,1$ with four copies of each molecule in the asymmetric unit (AU) (Table 1). Each TRIM2 RING protomer contacts concomitantly an adjacent (proximal) UBE2D1-Ub conjugate and the ubiquitin conjugated to the E2 molecule (distal) bound to the opposite RING (Fig. 2a). This arrangement stabilizes the E2-Ub closed conformation activated for ubiquitin transfer as observed in other E3/E2-Ub complexes, including TRIM25 RING/UBE2D1-Ub[31] and TRIM23 RING/UBE2D2-Ub[32]. The two dimeric TRIM2 RING/E2-Ub complexes in the AU are nearly identical with an rmsd between the two RING dimers of 0.24 Å, a maximum rms difference of 0.38 Å between four E2 molecules and equivalent ubiquitin orientation in all E2-Ub conjugates.

The RING dimer is assembled via a four-helix bundle formed by α-helices N-terminal (α1) and C-terminal (α3) to the core RING domain. These α-helices establish a hydrophobic interface involving the side chains of residues V12, I16, and F20, located in helix α1 of RING1 and the side chains of F82, L86, and V89 in the opposite RING2 α3 helix (Fig. 2b and Supplementary Fig. 3a). Furthermore, intra-protomer hydrophobic interactions between residues V13, L21 and L90 contribute to the interface integrity whilst residues V35 and L39 within the core RING domain form a 'hand-shake-type' interaction with the same residues in the opposite core (Fig. 2b). The arrangement of the TRIM2 RING dimer is reminiscent of other TRIM dimeric RING structures, such as the constitutive dimers TRIM32 (5FEY) and TRIM69 (6YXE) or those of TRIM25 (5FER) and TRIM5α (4TKP) that show minimal self-association in solution but crystallized as dimers in complex with E2-Ub conjugates (Fig. 2c)[31,33,34]. The calculated gain in solvation free energy of TRIM2 RING dimerization is −19.8 kcal/mol, a value larger than that calculated for the constitutive RING dimers of TRIM32 (−14.7 kcal/mol) and TRIM69 (−12.5 kcal/mol)[30,33]. TRIM2 RING dimerization is necessary for activity: L86 is buried in the core of the four-helix bundle (Fig. 2b) and, similarly to the V72R mutation in TRIM25 or I85R in TRIM32[31], its mutation to arginine greatly reduces catalytic activity in ubiquitin discharge assays (Fig. 3b).

In the RING/UBE2D1-Ub complex the E2-Ub interactions involve the classic ubiquitin I44 hydrophobic patch that, in tandem with V70, packs against UBE2D1 L104 in helix α1[35] (Supplementary Fig. 3b). Additional salt bridges between E2 residues D42 and D112 and ubiquitin K48 and R42, respectively, help maintain the E2-Ub closed conformation. E2/RING contacts involve UBE2D1 residues K4, K8, D12 and D16 in helix α1 and α1-β1 loop forming electrostatic interactions with the side chains of K18 and E28 of the RING domain (Supplementary Fig. 3c). Additional hydrophobic interactions are formed by the E2 side chains of F62 in the β3-β4 loop and P95 and A96 in the α2-α3 loop facing the TRIM2 side chains of I25 in the α1-β1 loop, Y50 in helix α2 and P61 and V62 in the β3-β4 loop.

**Table 1 | Data collection and refinement statistics**

| | TRIM2 RING/UBE2D1-Ub |
|---|---|
| Wavelength (Å) | 1.282 |
| Resolution range (Å) | 44.37–2.53 (2.62–2.53) |
| Space group | $P\,1\,21\,1$ |
| *Cell dimensions* | |
| a, b, c (Å) | 57.32, 123.7, 112.5 |
| α, β, γ (°) | 90, 102.8, 90 |
| Total reflections | 332463 (31614) |
| Unique reflections | 51031 (5126) |
| Multiplicity | 6.5 (6.2) |
| Completeness (%) | 99.79 (99.71) |
| Mean I/sigma(I) | 4.11 (0.38) |
| Wilson B-factor | 42.64 |
| R-meas | 0.2496 (2.283) |
| R-pim | 0.0972 (0.9172) |
| CC1/2 | 0.99 (0.537) |
| CC* | 0.998 (0.836) |
| Reflections used in refinement | 50947 (5111) |
| Reflections used for R-free | 2561 (271) |
| R-work | 0.2108 (0.3335) |
| R-free | 0.2556 (0.3758) |
| CC (work) | 0.957 (0.806) |
| CC (free) | 0.938 (0.711) |
| Number of non-hydrogen atoms | 9950 |
| Macromolecules | 9804 |
| Ligands | 8 |
| Solvent | 138 |
| Protein residues | 1234 |
| RMS (bonds) (Å) | 0.005 |
| RMS (angles) (°) | 0.84 |
| Ramachandran favored (%) | 98.02 |
| Ramachandran allowed (%) | 1.98 |
| Ramachandran outliers (%) | 0.00 |
| Rotamer outliers (%) | 0.63 |
| Clashscore | 7.39 |
| Average B-factor ($Å^2$) | 60.93 |
| Macromolecules ($Å^2$) | 61.12 |
| Ligands ($Å^2$) | 56.95 |
| Solvent ($Å^2$) | 47.30 |

Statistics for the highest-resolution shell are shown in parentheses.

Both RING protomers interact with the same ubiquitin molecule (Fig. 2a). In particular, the side chain of K33 of the proximal ubiquitin forms a salt bridge with D88 in the RING1 protomer whilst the backbone CO of Q19 at the N-terminal helix of RING2 forms a hydrogen bond with the amine group of K11 of the same ubiquitin molecule. In addition, RING2 K18 forms a hydrogen bond with the backbone carbonyl of ubiquitin G10, and the adjacent RING2 S24 side chain hydroxyl group is hydrogen bonded to the side chain of T9 of the ubiquitin (Fig. 3a). A similar network of interactions is observed in the complex of TRIM25 RING/UBE2D1-Ub where K65 of one RING forms a salt bridge with D32 of the proximal ubiquitin whilst K33 forms a hydrogen bond with N71 of the other RING[31]. Mutation of TRIM25 K65 to alanine greatly reduces catalytic activity. In contrast, mutation of TRIM2 K18, Q19, S24, N85, or D88 to alanine have no effect on the rate of ubiquitin lysine discharge (Fig. 3b), suggesting that single mutations within this RING/ubiquitin interface are ineffectual in compromising the stability of the E2-Ub closed conformation. On the contrary, mutation of the classic linchpin residue R64 in TRIM2 that contacts Q92 of the E2 and Q40 of the ubiquitin, ablates TRIM2 activity (Fig. 3a–c). Furthermore, F81 of the RING plays a crucial role in stabilizing the observed structural arrangement through a network of interactions. It is sandwiched between the peptide bond connecting E34-G35 of ubiquitin and L39 of the opposite RING, whilst adjacent H40 Nε2 hydrogen bonds with the backbone carbonyl of ubiquitin G35 (Fig. 3a). Mutation of F81 to alanine completely abrogates catalytic activity highlighting its central role (Fig. 3b, c).

### Solution properties of TRIM2 and TRIM3 RING domains

As we were not able to obtain crystals of the TRIM3 RING, we instead acquired solution structural information for both RING domains from Small-Angle X-ray scattering (SAXS) experiments (Supplementary Fig. 4a) to gain insight into the molecular basis for the reduced catalytic activity of TRIM3 compared to TRIM2. Details for data collection and analysis are reported in Supplementary Table 1. Whilst the Guinier analysis of the SAXS profiles results in similar values for the radii of gyration, both the Kratky plots and the distance distributions appear different for the two constructs (Fig. 4a). In particular, TRIM3 RING adopts a more elongated solution structure, with a $D_{max}$ of 74 Å compared to 66 Å for the TRIM2 RING. Moreover, the distinct uptrend of the curves in the Kratky plots at higher q-values shows that the RING of TRIM3 is more flexible than TRIM2 RING. TRIM3 RING has a smaller cross-sectional radius of gyration (Rc = 9.3 Å) than TRIM2 RING (Rc = 14 Å) and a SAXS-derived molecular weight shows it is monomeric in solution. The molecular weight of TRIM2 derived from the SAXS data is slightly smaller than the value for a dimeric species calculated from the primary sequence. Using the dimeric TRIM2 RING coordinates of the crystal structure to fit the X-ray scattering data results in a $\chi^2$ value of 14. This disagreement is likely the result, as suggested by our SEC-MALLS experiments, of TRIM2 RING not being a constitutive dimer in solution and to the disordered nature of the 10 N-terminal and 2 C-terminal residues for which there is no electronic density. Similarly, the theoretical scattering pattern calculated from the available AlphaFold2 coordinates for the monomeric RING domain of TRIM3 (residues 2–95) does not fit the experimental SAXS data ($\chi^2 = 4.4$) in line with the low model confidence for the prediction of the regions N- and C-terminal to the core RING domain[36].

We therefore applied a restrained molecular dynamics protocol, as implemented in Xplor-NIH, to obtain a more accurate description of the solution structure of the TRIM3 RING based on our SAXS data. We started with the available TRIM3 RING AlphaFold2 coordinates, considered those residues with a confidence score higher than 70 as rigid and allowed for changes in the conformation and orientation of the remaining residues. The best 10 conformers, selected based on their agreement with the experimental SAXS data ($\chi^2 = 1.34 \pm 0.04$), exhibit an extended solution structure for TRIM3 RING with an unfolded and dynamic N-terminal region (Fig. 4b). The conformer with the lowest $\chi^2$ value overlaps well with the best ab initio DAMMIF-derived envelope, which accommodates the unfolded N-terminal region of the RING in a protruding lobe (Fig. 4b). In summary, our X-ray scattering data show that in solution the RING domain of TRIM3 is a monomeric species and is consistent with a solution structure possessing an intact C-terminal α-helix (α3) but an unfolded and flexible N-terminal region.

### TRIM2 and TRIM3 interaction with E2 conjugating enzymes

Based on the observation that full-length TRIM3 manifests lower catalytic activity than TRIM2 in auto-ubiquitination assays (Fig. 1c and Supplementary Fig. 1) and in ubiquitin discharge assays, we wondered if the RING domain of TRIM3 was impaired in its ability to interact with an E2. We used NMR to monitor chemical shift perturbations produced in the $^1H$-$^{15}N$ HSQC spectrum of a $^{15}N$-labeled sample of UBE2D3 upon the addition of either TRIM3 or TRIM2 RING (Fig. 4c). Fast-intermediate exchange is observed for a number of amide

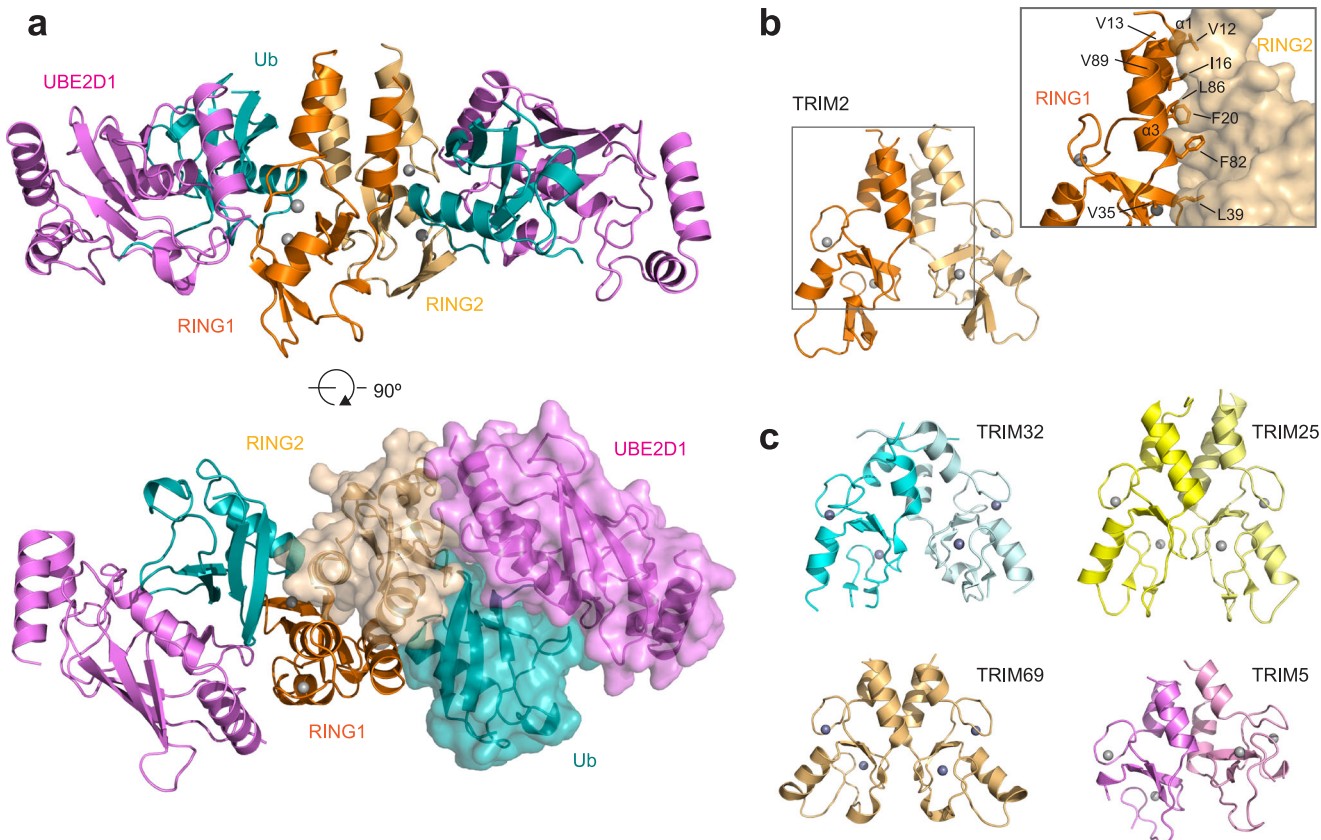

**Fig. 2 | Crystal structure of TRIM2 RING in complex with the UBE2D1-Ub conjugate. a** Cartoon representation of the crystal structure of the homodimeric RING of TRIM2 bound to two UBE2D1-Ub conjugates. The bottom panel is related to the upper one by a 90° rotation about the *x*-axis and shows one RING, E2 and Ub in surface representation. The RING domains are reported in orange (RING1) and light orange (RING2) whilst the E2 is in magenta and the ubiquitin in teal. The Zinc atoms in the RING domains are represented as gray spheres. **b** Details of the RING1-RING2 hydrophobic four-helix bundle dimerization interface (also see Supplementary Fig. 3a). **c** Cartoon representation of the homodimeric RING domain structures of TRIM32 (5FEY), TRIM25 (5FER), TRIM69 (6YXE) and TRIM5 (4TKP).

resonances in the E2 $^1$H-$^{15}$N HSQC spectrum with an associated overall line broadening particularly notable for the titration of UBE2D3 with TRIM2 RING as a consequence of the formation of the larger dimeric TRIM2 RING/E2 complex (55.6 kD). In both experiments, the addition of each ligand results in an identical chemical shift perturbation signature in the $^1$H-$^{15}$N HSQC spectrum of UBE2D3, consistent with equivalent residues experiencing the same changes in their chemical environment (Fig. 4d). Moreover, binding affinities obtained from fitting the chemical shift changes of a subset of resonances at different RING concentrations, indicate that the strength of the interaction between the RINGs and the E2 is of the same order of magnitude, with $K_d$ values of around 590 μM and 300 μM for TRIM2 and TRIM3 RING, respectively (Supplementary Fig. 4b). Mapping the resonance perturbations observed in UBE2D3 onto UBE2D1 in our TRIM2 RING/E2-Ub complex structure highlights a surface that overlaps with the RING-E2 interface (Fig. 4e). In particular, the surface encompasses helix α1 and the N-terminal side of α3 of the E2 directly in contact with the RING. Interestingly, the exposed perturbations extend to a surface on the E2 away from the RING binding site to both the surface that binds the ubiquitin in the closed conformation, made by helices α2 and α3, and the opposite site of the E2, clustering around residue S22 in the β1-β2 sheet. This latter area corresponds to the UBE2D family backside binding site for the allosteric ubiquitin and mutation of the serine residue to an arginine greatly reduce the rate of auto-ubiquitination in TRIM2 (Supplementary Fig. 4c)[37]. In both titration experiments, a number of E2 resonances promptly disappear upon addition of the E3. These resonances correspond to the amide protons of residues in the α2-α3 loop in direct contact with the RING and adjacent to the region

participating in the allosteric communication between the ubiquitin binding sites. These are likely residues experiencing large chemical shift changes and consequently extensive line broadening effects. Taken together, our NMR data show that the RING domains of TRIM2 and TRIM3 both bind the E2 in a similar fashion. Hence, the absence of activity observed for the TRIM3 RING is likely the result of a lack of RING self-association and ensuing inability to stabilize the E2-Ub closed conformation.

## TRIM2/3 phylogenetic analysis and rescue of TRIM3 RING activity

The lack of detectable catalytic activity of the TRIM3 RING is surprising given the high sequence similarity between the RING domains of TRIM2 and TRIM3 with 71% sequence identity and 86% similarity (BLOSUM62). TRIM2 and TRIM3 are coded by paralogous genes that arose by duplication in the common ancestor of jawed vertebrates (gnathostomes). In contrast, invertebrates, and other chordates including lampreys and hagfish, the earliest-diverging living vertebrate lineage, have a single TRIM2/3 gene (Supplementary Fig. 5a). Across species, the overall defining differences between TRIM2 and TRIM3 RING domains are restricted to: a) the N-terminus (TRIM2 residues 3–9, as reference); b) position 54, which in TRIM2 is always a histidine but in TRIM3 is a glutamine, and c) position 31 that in TRIM2, but not in TRIM3, is always a lysine (Fig. 5a). Nevertheless, some key residues in helices α1 and α3, part of the dimerization interface in TRIM2 RING, differ between the two proteins. To investigate if these non-conserved residues are the reason for the observed differences and to test if TRIM3 RING activity can be rescued we mutated a number of residues

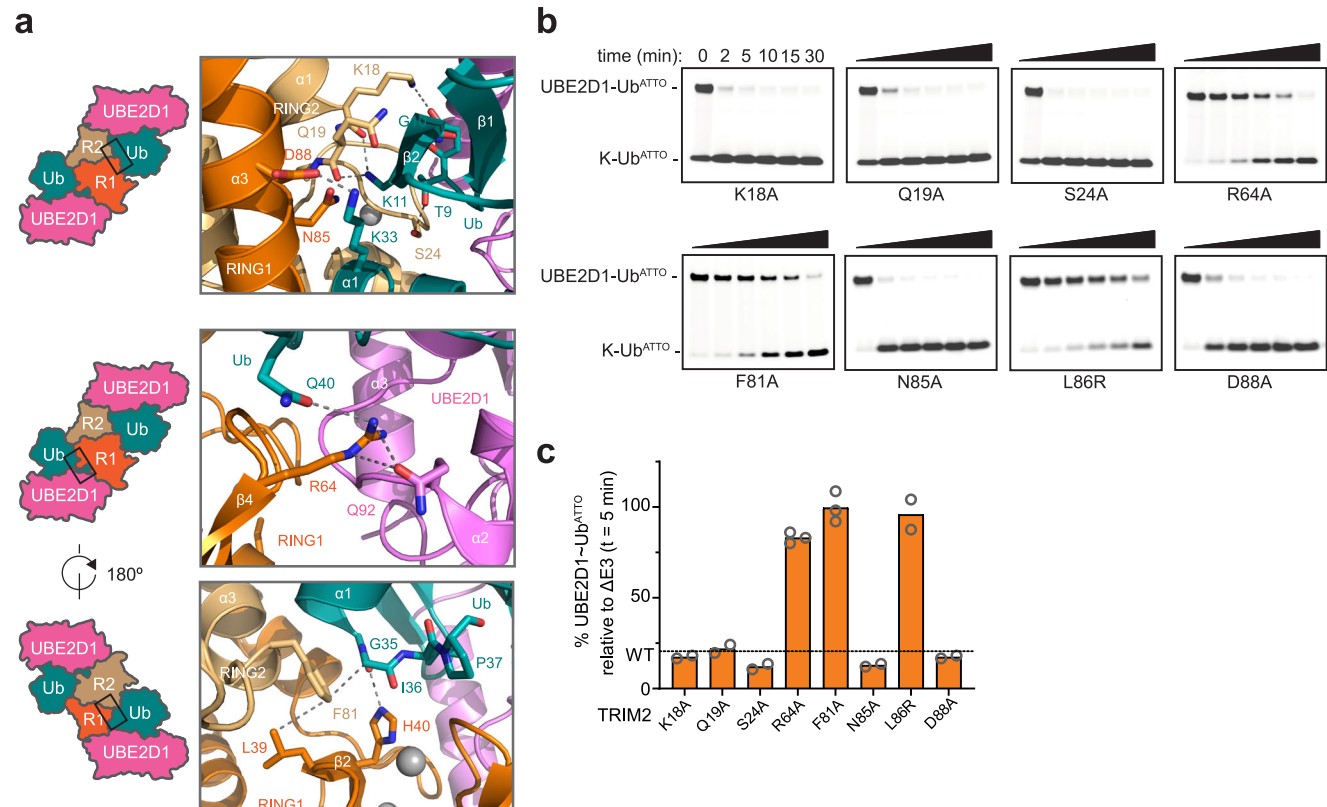

**Fig. 3 | TRIM2/E2-Ub interaction. a** Close-up of the interfaces between TRIM2 RING domain and the UBE2D1-Ub conjugate highlighted on a graphical illustration of the complex. Protein coloring is the same as in Fig. 2. **b** Lysine-discharge assay showing the disappearance of the band for the fluorescent labeled Ub^ATTO loaded E2 in the presence of TRIM2 RING mutant constructs. Reactions were carried over a 30-min time interval. **c** Quantification of the UBE2D1-Ub^ATTO band after 5-min reaction time reported as normalized intensity relative to the experiment in the absence of E3. The mean loss of E2-Ub for the WT protein is reported as a dotted line. The data are presented as mean values of at least two independent experiments ($n \geq 2$) for each RING mutant. Individual values are shown as gray circles. Source data is provided as a Source Data file.

in the regions N- and C-terminal to the core RING domain. TRIM2 V12 is positioned at the N-terminus of helix α1 and its side chain points towards the hydrophobic core of the four-helix bundle stabilizing the dimeric structure (Fig. 2b). In TRIM3 the equivalent residue is a glutamate (E11). Whilst the V12E mutation in TRIM2 is sufficient to abrogate activity, the reverse E11V mutant in TRIM3 RING remains inactive (Fig. 5b). V89 in helix α3 of TRIM2 interacts with V12, in TRIM3 the equivalent residue is A88. Mutation of A88 into a valine alone or in combination with the E11V mutation does not restore TRIM3 activity, whilst the V89A mutation in TRIM2 reduces its activity in discharge assays, and the double mutant V12E/V89A almost entirely abrogates TRIM2 activity (Fig. 5b). Proline 14 in TRIM3 corresponds to a glutamine (Q15) in TRIM2 and is positioned at the center of helix α1. Proline residues are generally found at the beginning or end of a helix as they tend to disrupt regular secondary structures and indeed, the P14Q mutation in TRIM3 partially restores activity (Fig. 5b) although only in combination with the single E11V mutation, but not when the A88V mutation is also present. Interestingly, the RING domain of TRIM71, another TRIM class VII member, also lacks apparent E3 catalytic activity in lysine discharge assays (Fig. 5c). This is likely due to the presence of a proline residue (P5) in the N-terminal region and a large serine-glycine rich insertion that prevents RING dimerization. The disruption of α1, and consequent loss of E3 activity, caused by the V11E/R13Q/Q14P substitutions we describe here for human TRIM3 RING, is a feature only observed in placental mammals (Fig. 5a and Supplementary Fig. 5a). From fish to marsupials, the α1 helix of TRIM3 is predicted to be intact (AlphaFold2). Intriguingly, avian TRIM3 and human TRIM2 RINGs have conserved sequences in the α1 and α3 helical regions

(Fig. 5a) and indeed *Gallus gallus* (chicken) TRIM3 RING discharges ubiquitin from a pre-loaded UBE2D1 as efficiently as human TRIM2 RING (Fig. 5c). Moreover, its AlphaFold2 predicted structure shows an intact N-terminal helix and SEC-MALLS data show that *Gg*TRIM3 RING has a strong self-association tendency similarly to the human TRIM2 RING (Supplementary Fig. 5b). Taken together these results suggest that single amino acid changes are not sufficient to restore full TRIM3 activity and that both the integrity of the N-terminal α1 helix and its complementarity with the C-terminal α3 helix are necessary to stabilize a RING dimer and, consequently, the active E2-Ub conformation.

We hypothesized that the absence of human TRIM3 RING activity is primarily due to lack of dimerization but not an intrinsic property of the RING, explaining why full length TRIM3 shows catalytic activity, albeit severely reduced. To test this model, we produced tandem constructs containing either two consecutive TRIM3 RING (T3R-T3R) domains or TRIM3 RING/TRIM2 RING (T3R-T2R) connected by a single serine residue. Strikingly, both constructs were as efficient in lysine discharge experiments as the TRIM2 RING domain (Figs. 1e, 5c). Importantly, the activity of the fusion constructs is not the result of TRIM2 RING trans-dimerization as both T3R-T3R and T3R-T2R do not undergo further self-association as shown by SEC-MALLS experiments (Supplementary Fig. 6a). Furthermore, TRIM3 RING fused to TRIM2 RING is directly involved in the stabilization of the E2-Ub closed conformation as mutant T3R-T2R (F81A) is still efficient in catalyzing the discharge of the ubiquitin molecule onto free solution lysine (Fig. 5c and Supplementary Fig. 6b). Similarly, a construct where the segments N- and C-terminal to the TRIM3 core RING domain were substituted by the equivalent TRIM2 RING α-helices (T2αs-T3Rcore) has the same

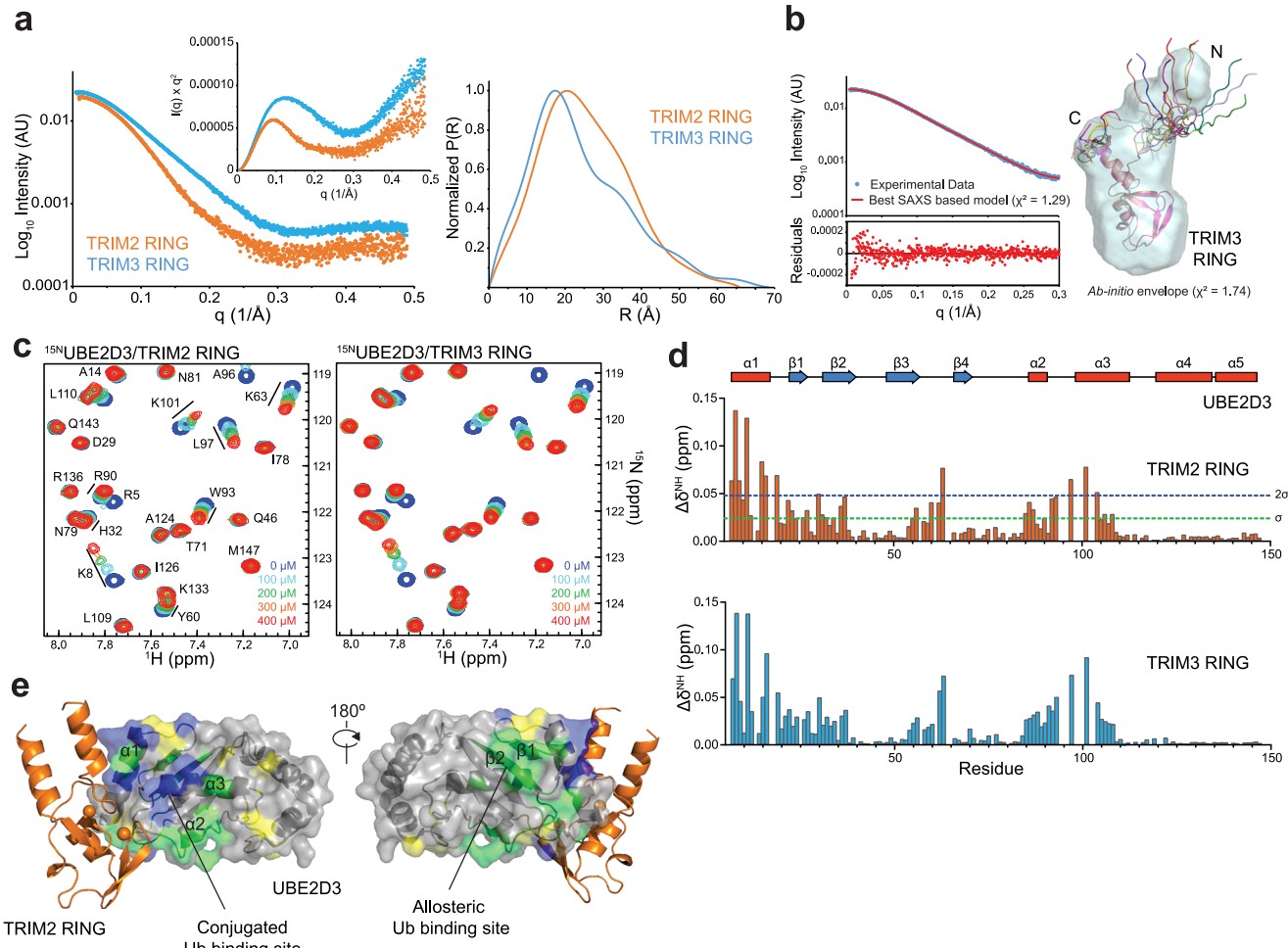

**Fig. 4 | Small-Angle Xray scattering and NMR spectroscopy. a** X-ray scattering profiles, Kratky plots and normalized pair-distribution functions P(R) for the RING domains of TRIM2 (orange) and TRIM3 (cyan). **b** *Left panel*: agreement of the experimental SAXS intensities of TRIM3 RING (cyan) with those calculated by FoXS for the representative conformer (lowest $\chi^2$) derived from molecular dynamics (red). *Right panel*: overlap of the best DAMMIF ab initio molecular envelope and the ensemble of the 10 lowest $\chi^2$ conformers obtained by Xplor-NIH. The representative conformer has been overlapped to the ab initio envelope using SUPCOMB and the remaining nine protomers have been aligned to the representative one using residues 19–88 (treated as a rigid body in the modeling protocol). **c** Details of the

2D $^1$H-$^{15}$N HSQC spectra of UBE2D3 titrated with the RING domain of TRIM2 and TRIM3. Spectra at different ligand concentrations are plotted at the same contour level. **d** Plot of the chemical shift perturbations ($\Delta\delta^{NH}$) *vs* residue number in the NMR spectra of $^{15}$N-labeled UBE2D3 in the presence of two molar equivalents (400 µM) of the RING domains of TRIM2 (orange) and TRIM3 (cyan). Secondary structure of UBE2D3 is reported as a function of residue number. **e** The residues of UBE2D3 with perturbed $^1$H-$^{15}$N chemical shifts above 1σ (green) and 2σ (blue) are represented on the structure of TRIM2 RING bound to UBE2D1 in which the E2 has been replaced by UBE2D3 (5EGG). Proline residues on the E2 are colored in yellow. Source data is provided as a Source Data file.

---

SEC-MALLS profile as wild-type TRIM2 RING at the same concentration (Supplementary Fig. 6a) and shows the same level of catalytic activity (Fig. 5c). These experiments demonstrate that TRIM3 RING activity can be fully rescued by inducing homo- and even heterodimerization and highlight that the integrity of the N-terminal helix is crucial for RING activity by promoting RING dimerization.

**TRIM2 and TRIM3 interact in mammalian cells**

To investigate if rescue of TRIM3 activity upon association with TRIM2 may have a physiological role, we tested if TRIM2 and TRIM3 interact in cells. Indeed, TRIM2 and TRIM3 co-immunoprecipitate when over-expressed in HEK293T cells (Fig. 6a and Supplementary Fig. 7a). To investigate if TRIM2 and TRIM3 also interact at endogenous levels we first tested their individual expression levels in multiple human cell lines (Supplementary Fig. 7b). Given the high degree of sequence identity between the two TRIM proteins, we first validated a panel of antibodies to ensure they don't cross-react (Supplementary Fig. 7c). Co-immunoprecipitation experiments from a brain-derived cell line (LN229) show that endogenous TRIM2 and TRIM3 can bind each other

in a physiological context (Fig. 6c), Furthermore, they co-localize (Pearson's R value 0.73) at lamellipodia-like protrusions at the cell periphery, consistent with previous reports indicating their roles in regulating cytoskeletal proteins (Fig. 6d)[17,19,22–24].

To map which domains are important for the interaction between TRIM2 and TRIM3, a series of truncation mutants were expressed in HEK293T cells (Fig. 6e–g). Deletion of some domains of TRIM3, but not TRIM2, destabilized expression in HEK293T cells, which was partially rescued by the use of a proteasome inhibitor. Comparison of immunoprecipitated protein to input revealed that the interaction of TRIM2 and TRIM3 was abrogated by the loss of their filamin or coiled-coil domains, respectively. This suggests that TRIM2 and TRIM3 interact in an asymmetric manner, where the filamin domain of TRIM2 contacts the coiled-coil region of TRIM3. However, in contrast to some other TRIM proteins, the B-box domain does not contribute to higher order association.

Intriguingly, immunoprecipitated TRIM3 becomes poly-ubiquitinated during in vitro ubiquitination assays when TRIM2 is co-expressed, but not when expressed alone, indicating that TRIM3 either

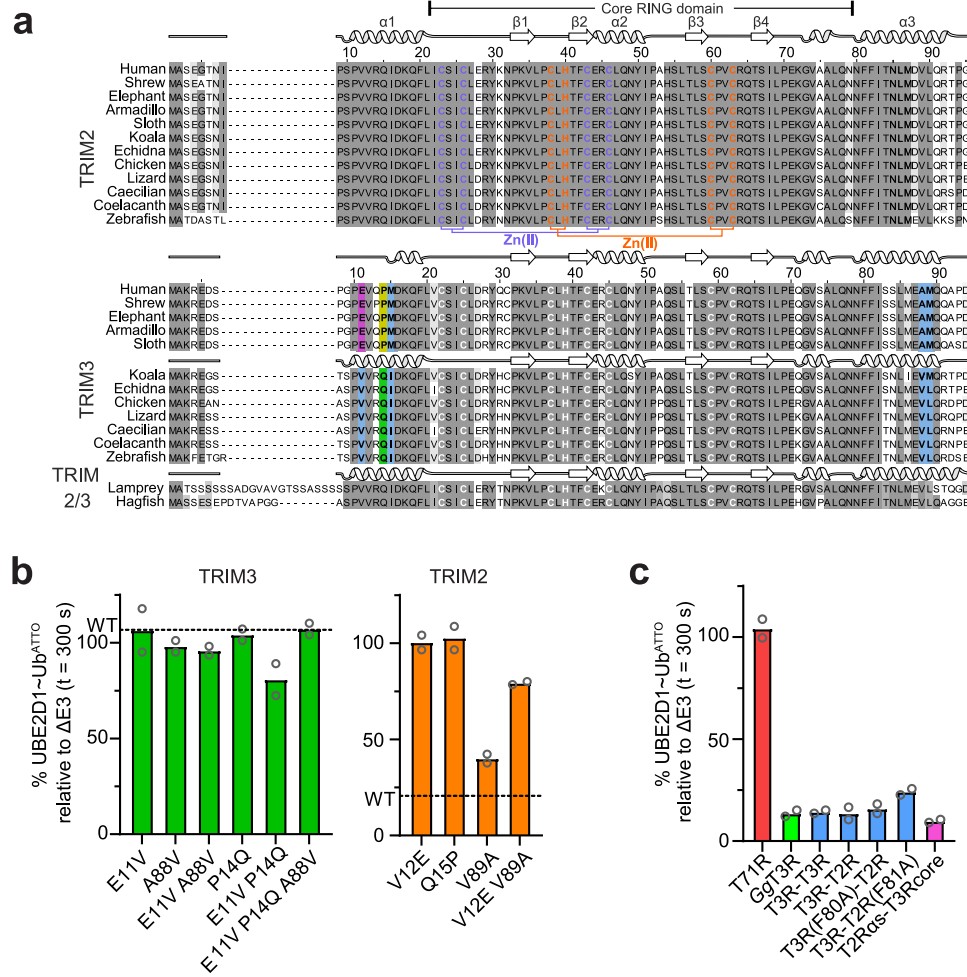

**Fig. 5 | TRIM2/3 sequence conservation and rescuing activity of TRIM3.**
**a** Multiple sequence alignments of TRIM2 and TRIM3 RING domains across different species. The secondary structures reported above the alignments are those of the crystal structure for TRIM2 RING in complex with UBE2D1-Ub and those predicted for TRIM3 RING from different species by AlphaFold2. **b** Quantification of the UBE2D1-Ub^ATTO band after 5-min reaction time reported as normalized intensity relative to the experiment in the absence of E3 for mutant constructs of TRIM3 and

TRIM2. The mean loss of E2-Ub for TRIM2 and TRIM3 RING WT are reported as dotted lines in the relative plots **c** UBE2D1-Ub^ATTO discharge activity of TRIM71 RING (T71R), *G.gallus* TRIM3 RING, TRIM2/TRIM3 tandem constructs and T2Rαs-T3Rcore chimera. The data are presented as mean values of two independent experiments ($n = 2$) for each construct. Individual values are shown as gray circles. Source data is provided as a Source Data file.

is a substrate of TRIM2, or that association with TRIM2 promotes TRIM3 ligase activity (Fig. 6a, b). Conversely, the ubiquitin ligase activity of TRIM2 appears reduced in complex with TRIM3, suggesting that TRIM3 may bind TRIM2 in order to regulate ubiquitination activity. Further studies are required to understand the physiological implications of this observation in detail.

## Discussion
TRIM proteins constitute a protein family with highly diverse functions but a common architectural feature, the TRIM or RBCC motif, which encompasses a conserved RING domain, which is best known for its ubiquitin transfer activity. While only a limited number of TRIM proteins have been structurally characterized, it appears that a common feature of their RING domains is the requirement for dimerization for detectable E3 ligase activity. Self-association may occur either in a constitutive manner as observed for TRIM32[31] or TRIM69[33], or reversibly where weak self-association is strengthened and stabilized upon interaction with the E2-Ub conjugate[30], and in some cases through higher order clustering (e.g., TRIM5α or TRIM21)[38–40]. In contrast, TRIM proteins with RING domains that lack catalytic activity, such as TRIM28, contain strictly monomeric RINGs and activity cannot be restored by enforced dimerization[41]. Nevertheless, some non-TRIM

RINGs are active as monomers and for these stabilization of the closed E2-Ub conformer is attained by features outside the RING domain, such as by a phosphotyrosine in Cbl-b[42] or through binding a regulatory ubiquitin molecule in Ark2C[43]. However, no such mechanism has hitherto been described for TRIM E3 ligases.

Here we present a detailed analysis of two members of class VII TRIM proteins, TRIM2 and TRIM3. Despite their high sequence similarity, they display very different enzymatic properties. TRIM2 is an active E3 ligase with a RING domain with a strong tendency to dimerize whilst TRIM3 has very reduced catalytic activity compared to TRIM2 with a RING domain that in isolation is monomeric and catalytically inactive. We can exclude that the lack of activity of TRIM3 is due to autoinhibitory interactions as observed in TRIM21 where the E2-binding site of the RING is occluded by the B-box that is displaced upon phosphorylation, as in this case the isolated RING should show uninhibited levels of activity[44]. A comparison between the crystal structure of the TRIM2 RING/UBE2D1-Ub complex and a model for the solution structure of the TRIM3 RING based on SAXS data instead shows that dimerization of TRIM3 RING is impaired due to an unfolded N-terminal segment which prevents the formation of a four-helix bundle by regions N- and C-terminal to the core RING domain. Such a four-helix bundle is present in other active TRIM RINGs and is

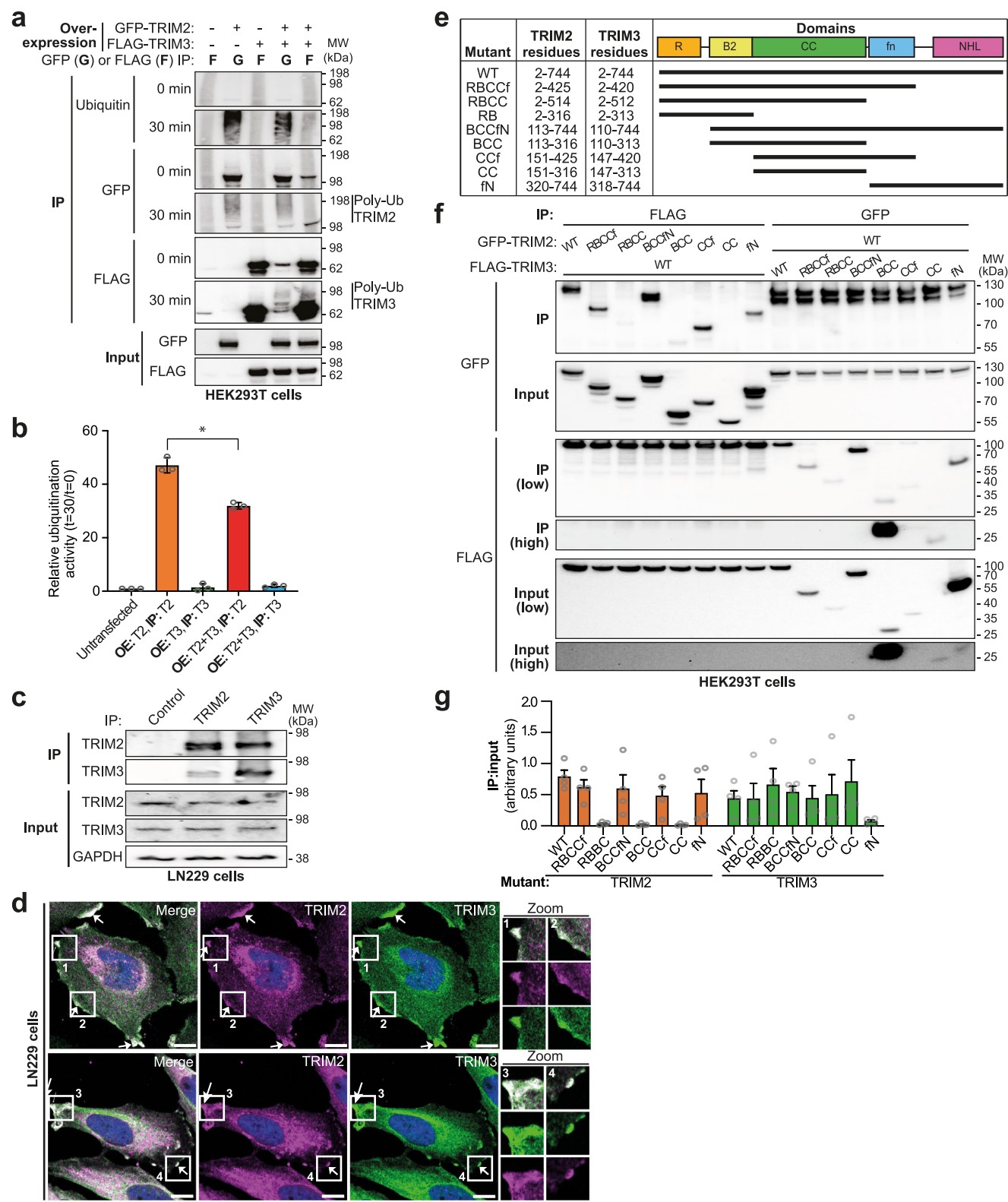

necessary for the stabilization of the activated E2-Ub conformation[30]. Remarkably, self-association and hence activity can be restored upon enforced dimerization by covalently linking the RINGs to form either homodimers or TRIM2-TRIM3 heterodimers or by substituting the N- and C-terminal helices with those of TRIM2. This shows that the core RING domain contains all features required for mediating ubiquitin transfer, an interpretation supported by its high sequence conservation. Furthermore, this is the

first demonstration of ligase activity in a hetero-dimerized TRIM RING in which one RING is inactive on its own, highlighting that association between different TRIM proteins may be a mechanism to regulate their catalytic activity.

Taken together, the data presented here highlight that minor changes in the regions N- and C-terminal to the core RING domain can have severe effects on E3 ligase activity of TRIM proteins and lead us to speculate that regulation of RING homo- or potentially hetero-

**Fig. 6 | Interaction of TRIM2 and TRIM3 in cells. a** in vitro ubiquitination assay reaction using TRIM2 and/or TRIM3 overexpressed (OE) in HEK293T cells, isolated by immunoprecipitation (IP) using their respective tags as indicated, followed by western blotting using the indicated antibodies to analyze whole cell lysate inputs and before (0 min) or after a 30-min reaction (also see Supplementary Fig. 7a). **b** Quantification of ubiquitin signal from western blotting analysis of experiment described in part a, given as a ratio of signal after 30-min reaction time relative to starting material. The values are reported as mean of three independent experiments ($n = 3$), gray circles mark individual values, error bars represent mean ± SEM, asterisk represents a two-sided Student's $t$ test $p = 0.0011$. **c** Co-immunoprecipitation of endogenous TRIM2 and TRIM3 proteins from LN229 glioblastoma cells, alongside a control IP using FLAG antibody, analyzed by western blotting using the indicated antibodies, representative of three independent experiments. **d** Representative images from three independent immuno-fluorescence experiments showing endogenous TRIM2 (magenta) and TRIM3 (green) co-localization in LN229 glioblastoma cells (nuclei stained with DAPI (blue), scale bar represents 10 μm). **e** Table representing different truncation mutants of TRIM2 and TRIM3. **f** Co-immunoprecipitation of overexpressed truncation mutants of TRIM2 and TRIM3 by their respective tags from HEK293T cells pre-treated with 10 μM MG132 6 h, with labels corresponding to the diagram above in e, analyzed by western blotting against the indicated antibodies. **g** Quantification of GFP or FLAG western blotting for each of the truncation mutants of TRIM2 (orange) and TRIM3 (green), respectively, given as a ratio of IP to input. The values are reported as mean of four independent experiments ($n = 4$), gray circles mark individual values, error bars represent mean ± SEM. Source data is provided as a Source Data file.

dimerization may be a mechanism to fine-tune catalytic activity in response to different stimuli, and to prevent unwarranted ubiquitination of substrates that may have detrimental consequences.

It is intriguing that despite being so highly conserved, TRIM2 and TRIM3, show such differences in their activity, and we can only speculate as to the driver of the observed strong attenuation of TRIM3 ligase activity in placentals. It is possible that, following the gene duplication in the last common ancestor of jawed vertebrate ca. 550 mya, TRIM2 and TRIM3 started experiencing separate evolutionary pressures due to their different genomic context, eventually resulting in subfunctionalisation and/or neofunctionalisation. Originally, and this appears to be the case for most vertebrate lineages, the evolutionary pressure was for both to remain active Ub ligases. When placental mammals originated (ca. 150 mya) changing needs or pressures may have relaxed this constraint to allow the evolution of a ligase-independent activity in TRIM3. Perhaps in placentals TRIM3 outcompetes other TRIMs for substrate binding and thus protects them from ubiquitination, or, alternatively, TRIM3 may function as a scaffold to bring other proteins/RNA together via its NHL domain. Conversely, TRIM3 could work in conjunction with TRIM2 or other TRIM proteins. TRIM2 and TRIM3 co-express in the same cell types, interact in yeast-two-hybrid assays[45], and we now show also interact in mammalian cells via their respective filamin and coiled-coil domains (Fig. 6). Indeed, co-expression of TRIM3 with TRIM2 was seen to reduce TRIM2 activity (Fig. 6a, b), suggesting it might be a regulator of TRIM2 function. An interesting observation is that the time when the evolutionary paths of placental TRIM2 and TRIM3 started to diverge coincides with the emergence of a new section of the brain with specific functions. TRIM2 and TRIM3 are both expressed in the brain but their expression levels in different cell types are not equivalent: TRIM2 is mainly present in the corpus callosum whilst TRIM3 is mostly found in the cerebellum (Human Protein Atlas)[46]. The corpus callosum plays a role in spatial and sensory coordination by connecting the two brain hemispheres through a large fiber tract and it is an exclusive feature of the placental brain[47]. This raises the fascinating hypothesis that perhaps the divergence of TRIM2 and TRIM3 contributed to the acquisition of higher functions of an evolving brain.

## Methods

### Proteins expression and purification and UBE2D1-Ub preparation

Cloning, expression, and purification of the E1 (Ube1) and E2 (UBE2D1) enzymes have been described previously[31]. Human TRIM2 constructs RING (2–95), RB2 (2–156), RBCC (2–316), TRIM3 constructs RING (2–95), RB2 (2–153), RBCC (2–313) and TRIM71 RING (2–128) were cloned by Gibson Assembly into pET49b vector as HRV 3C protease cleavable N-terminal His$_6$-fusion proteins. The TRIM3-TRIM3 and TRIM3-TRIM2 tandem constructs were produced by linking two copies of each RING domain by a single serine residue whilst a chimera protein was created by adding the N- and C-terminal α-helices of TRIM2 (2–16/84–95) to the core

RING domain of TRIM3 (16–82). These constructs were synthesized as gblocks (IDT) and ligated by Gibson Assembly into pET49b vector as HRV 3C protease cleavable N-terminal His$_6$-fusion proteins. Mutations were introduced using a Quickchange-based protocol. All constructs were verified by DNA sequencing.

All proteins were expressed in BL21 (DE3) *Escherichia coli* cells. Cells were grown in LB supplemented with 200 μM ZnCl$_2$ at 37 °C until the OD reached 0.6–0.8 then induced with 0.5 mM IPTG at 18 °C for 16 h. The proteins were purified by Ni-NTA affinity chromatography followed size-exclusion chromatography after His$_6$-tag cleavage by HRV 3 C protease. Samples were placed in the final buffer of 20 mM HEPES pH 7.5, 150 mM NaCl and 0.5 mM TCEP.

Full length TRIM2 (1-744) and TRIM3 (1-744) were cloned with an uncleavable C-terminal His$_6$-tag into the baculovirus transfer vector pACEBac1. The resulting bacmid was used to transfect Sf9 cells. The cells were cultured at 28 °C for 72 h in SF900 III serum-free medium (Invitrogen). Expression of the TRIM proteins was obtained by infecting the Sf9 cells at $2 \times 10^6$ cells/mL density with 1 mL of high-titer TRIM-baculovirus for 72 h. The proteins were purified using a three-step protocol consisting of an immobilized-metal affinity chromatography step, followed by ion-exchange (Resource Q, GE-Healthcare Life Sciences and pH 8) and size-exclusion (Superdex S200 XK16/60) in 25 mM HEPES pH 8.0, 300 mM NaCl and 0.5 mM TCEP. Ubiquitin from bovine erythrocytes (Sigma, U6253), used in all ubiquitination assays, was solubilized in 20 mM HEPES pH 7.5, 150 mM NaCl and 0.5 mM TCEP and purified by size-exclusion in the same buffer. His$_6$-M1C-ubiquitin labeled with ATTO 647 N maleimide and UBE2D1-Ub$^{\text{ATTO}}$ were prepared with a procedure described previously[48,49]. For crystallography, TEV-protease cleavable N-terminal His$_6$-tagged human ubiquitin (1–76) was cloned into a pET21. As a result, the construct contains the tetrapeptide GAMG at the N-terminus after treatment with the protease. Isopeptide-linked UBE2D1-Ub (S22R, C85K) was prepared as previously described[50].

### Mammalian cell culture

HEK293T, HeLa, MEF, HT29, and U2OS cells were cultured in Dulbecco's modified eagle medium (ThermoFisher, 41966-029), supplemented with 10% FBS (ThermoFisher, 10270106) and 100 U/ml penicillin/streptomycin (Gibco, ThermoFisher Scientific), at 10% CO$_2$ at 37 °C. LN229, SF539, U87, U251, A172, and ST88-14 cells were cultured under the same conditions, but supplemented with a further 2 mM L-glutamine (ThermoFisher, 25030081).

### Mammalian endogenous TRIM2 and TRIM3 immunoprecipitation and immunofluorescence

For immunoprecipitation, LN229 cells were cultured to 90% confluency in 10 cm dishes before the addition of 1 ml of lysis buffer (0.5% IGEPAL, 150 mM NaCl, 50 mM Tris-HCl pH 7.5, 5 mM MgCl$_2$, protease inhibitors (Merck, 4693159001) and centrifugation 14,500 × $g$, 15 min, 4 °C. After a sample of the input was

taken, lysates were subjected to end-on rotation for 2 h at 4 °C with 2.5 µl of either anti-TRIM2 (Protein Tech, 16819925/67342-1-IG-150UL) or anti-TRIM3 (Abcam, ab111840) with 10 µl Protein A/G beads (Pierce, 88802). The resulting pull downs were washed three times with lysis buffer before the addition of 2X LDS sample buffer. Samples were then analyzed by western blotting against TRIM2 (Protein Tech, 16819925/67342-1-IG-150UL, 1:500), TRIM3 (Abcam, ab111840, 1:500), or GAPDH (Millipore, MAB374, 1:2,000) overnight at 4 °C, followed by anti-mouse-HRP or anti-rabbit-HRP secondary antibody (Dako, P0447 and P0399, 1:2,000) 1 h RT. Amersham detection reagents (RPN2106) were added then blots imaged using BioRad ChemiDoc and analyzed in ImageLab.

For immunofluorescence, LN229 cells were cultured to 75% confluency on 22 mm coverslips (VWR, 631-1582) in 6-well plates before fixation with 4% paraformaldehyde (ThermoFisher, 15670799) 30 min RT, permeabilization with 0.1% Triton X-100 (Sigma) in PBS 5 min RT, and blocking with 1% BSA (Sigma, A9647) in PBS 30 min RT. Samples were incubated with primary antibodies (TRIM2 (Protein Tech, 16819925, 1:50) and TRIM3 (Abcam, ab111840, 1:50)) in blocking buffer overnight at 4 °C, which were then detected with anti-mouse-Alexa594 or anti-rabbit-Alexa488 (ThermoFisher, A11032 and A11008) 1 h RT, and finally stained with 1 µg/ml DAPI (Merck, D9542) in PBS before mounting on slides with ProLong Gold Mountant (ThermoFisher, P10144). Slides were imaged on a Zeiss Invert880 confocal microscope under oil with 63X lens, taking $14 \times 0.36$ µm z-slices per field of view.

### Mammalian TRIM2 and TRIM3 overexpression and immunoprecipitation

For protein expression, 6 cm dishes of HEK293T cells were transfected for 4 h in OptiMEM (ThermoFisher, 31985062) and Lipofectamine 3000 (ThermoFisher, L3000008), as per the manufacturer's instructions, with pcDNA3.1-FLAG-TRIM3 (full length protein with C-terminal tag) and ptCMV-EGFP-TRIM2 (full length protein with N-terminal tag). Truncation TRIM2 and TRIM3 constructs were cloned into ptCMV-EGFP (N-terminal tag) and pcDNA3.1-FLAG (N-terminal tag), respectively and transfected in HEK293T cells similarly to full length constructs. Following transfection, OptiMEM was exchanged for normal growth medium for 24 h. Cells were then washed once with ice cold PBS and lysed in 500 µl lysis buffer (0.5% IGEPAL, 150 mM NaCl, 50 mM Tris-HCl pH 7.5, 5 mM MgCl₂, protease inhibitors (Merck, 4693159001) and centrifugation $14,500 \times g$, 15 min, 4 °C. In the case of overexpression of truncation mutants, cells were then treated with 10 µM MG132 6 h before lysis. After a sample of the input was taken, lysates were subjected to end-on rotation at 4 °C with either 2 µl anti-GFP (Roche, 11814460001) 2 h followed by 1 h Protein A/G beads (Pierce, 88802), or 10 µl FLAG M2 magnetic beads (Sigma, M8823) 2 h. The resulting pull downs were washed three times with lysis buffer then, where indicated, used in an in vitro ubiquitination assay, as described below.

### In vitro ubiquitination and ubiquitin lysine-discharge assays

For in vitro ubiquitination assays TRIM2 and TRIM3 constructs, at a final concentration of 4 µM, were incubated with 0.5 µM E1, 2.5 µM E2 (UBE2D1), 50 µM ubiquitin, 1 µM ubiquitin$^{ATTO}$, and 3 mM ATP in 50 mM HEPES pH 7.5, 150 mM NaCl and 20 mM MgCl₂. Reactions were incubated at 25 °C and each time point was quenched by addition of 2X LDS sample buffer supplemented with 0.5 M DTT and flash frozen in liquid N₂. The gels were scanned with a LI-COR CLx scanner.

For in vitro ubiquitination assays using proteins immunoprecipitated from mammalian cells, TRIM2 and TRIM3 were isolated from 6 cm dishes of 90% confluent HEK293T cells and incubated with 2 µM E1, 2 µM E2 (UBE2D1), 50 µM ubiquitin, and 3 mM ATP in 50 mM HEPES pH 7.5, 150 mM NaCl and 20 mM MgCl₂. Reactions were incubated at

25 °C for 30 min before the addition of 2X LDS sample buffer supplemented with 0.5 M DTT and analysis by western blotting against GFP (Roche, 11814460001, 1:1,000), FLAG (Merck, A8592, 1:10,000), or ubiquitin (Invitrogen, 13-1600, 1:1,000), which were detected by anti-mouse-HRP secondary antibody (Dako, P0447, 1:2,000). Blots were imaged using BioRad ChemiDoc and analyzed in ImageLab.

Ubiquitin lysine-discharge assays were performed with 1 µM pre-charged UBE2D1-Ub$^{ATTO}$, 4 µM TRIM constructs and 20 mM L-lysine in 50 mM HEPES pH 7.5 and 150 mM NaCl. Reactions were incubated at 25 °C and time points were quenched with the addition of 2X LDS sample buffer and flash frozen in liquid N₂. For quantification, the gels were scanned with a LI-COR Odyssey CLx scanner, and the bands of the E2-Ub$^{ATTO}$ were integrated using the ImageStudio software package (LI-COR). The scans were converted in greyscale. All the experiments were performed in duplicates or triplicates. The data were plotted using GraphPad.

E2scan (version 2, Ubiquigent) assays of TRIM2 and TRIM3 were executed according to the manufacturer protocol with 0.3 µM E1, 2.5 µM E2 (UBE2D1), 1 µM E3, 100 µM ubiquitin, and 2 mM ATP in 25 mM HEPES pH 7.5, 150 mM NaCl and 20 mM MgCl₂. Following SDS-PAGE, 1:1000 dilution mouse anti-ubiquitin antibody (Invitrogen, 13-1600) was used to detect mono- or poly-ubiquitination. A secondary antibody labeled with IRDye 800CW (LI-COR, goat anti-mouse IgG, 926-32210) was used at 1:8000 dilutions. The gel bands were visualized with the Odyssey CLx imaging system (LI-COR) and converted to greyscale for illustration.

### SEC-MALLS

Analytical SEC-MALLS profiles were recorded at 16 angles using a DAWN-HELEOS-II laser photometer (Wyatt Technology) and differential refractometer (Optilab TrEX) equipped with a Peltier temperature-regulated flow cell maintained at 25 °C (Wyatt Technology). 100 µl samples of purified proteins at multiple concentrations were applied to a Superdex 75 or 200 10/300 GL increase column (GE Healthcare) equilibrated with 20 mM HEPES pH 7.5, 150 mM NaCl, 0.5 mM TCEP, and 3 mM NaN₃ at a flow rate of 1 ml/min. Data were analyzed using ASTRA 6.1 (Wyatt Technology).

### Crystallization, data collection, phasing and refinement

TRIM2 RING and ubiquitin-conjugated UBE2D1 (S22R, C85K), both in 20 mM HEPES pH 7.5, 150 mM NaCl, 0.5 mM TCEP, were mixed at 215 µM concentration in a 1:1 ratio. Commercially available sitting drop crystallization screens were dispensed at 20 °C using an automated Mosquito machine (TTP Labtech). Crystals grew from a 100 nL protein solution plus 100 nL reservoir in 0.1 M HEPES pH 7.5 and 10% (w/v) PEG 8 K as precipitant. For X-ray data acquisition, crystals were cryoprotected with mother liquor containing 25% ethylene glycol. Diffraction data were collected on beamline IO3 ($\lambda = 1.282$ Å) at the Diamond Light Source (Oxford, UK), processed using DIALS[51] and merged and scaled using AIMLESS[52]. The structure of the complex was solved by molecular replacement using the RING domain and the E2-Ub conjugate from the structure of TRIM25/UBE2D1-Ub complex[31] (5FEY) as templates in Phenix Phaser[53]. Models were iteratively improved by manual building in Coot[54] and refined using Refmac[55] and Phenix[53]. Coordinates and structure factors are deposited in the Protein Data Bank under accession code 7ZJ3. Further details on data collection and refinement statistics are summarized in Table 1.

### Small-Angle Xray Scattering and TRIM3 RING structure modeling

SAXS data were collected at the SWING beamline at SOLEIL (GIF-sur-YVETTE CEDEX, France). The purified TRIM RING constructs, at 10 mg/ml (ca. 1 mM), were injected onto a Bio SEC-3 100 Å Agilent column and eluted at a flow rate of 0.2 ml/min at 15 °C. Frames were

collected continuously during the fractionation of the proteins. Frames collected before the void volume were averaged and subtracted from the signal of the elution profile to account for background scattering. Data reduction, subtraction, and averaging were performed using the software FOXTROT (SOLEIL). The scattering curves were analyzed using the package ATSAS[56] and reported as a function of the angular momentum transfer q = 4π/λ sinθ, where 2θ is the diffraction angle and λ the wavelength of the incident beam. Values of the cross-sectional radius of gyration were obtained with SCATTER[57]. Comparison with crystal and AlphaFold2 structures and the experimental scattering profiles was done with FoXS[58]. Low-resolution three-dimensional ab initio models for the TRIM3 RING molecular envelope was generated by the program DAMMIF[59] and overlapped to the NMR-derived structures using SUPCOMB[60]. The SAXS-derived dummy atom models were rendered with the PyMOL molecular graphics system. Further details on data collection and statistics are summarized in Supplementary Table 1.

TRIM3 RING solution structure modeling based on SAXS data was carried out using the available AlphaFold2[36] structure and an Xplor-NIH[61] simulated annealing protocol. During the calculation residues 19–88, with confidence interval higher than 70, were treated as a rigid body. The tripeptide Gly-Pro-Gly, residual form the 3C-protease cleavage, was added to the RING N-terminus, and these together with the residues 2–18 and 89–95 were allowed to move during the molecular dynamics steps. We generated different conformers using a protocol consisting of high-temperature torsion-angle dynamics at 3000 K, followed by simulated annealing from 3000 K to 25 K in 12.5 K increments and a final gradient minimization in torsion-angle space. In addition to the experimental SAXS-derived force field, knowledge-based energy terms, such as torsion angle potential derived from conformational databases[62], backbone hydrogen bond potential[63] and standard Xplor-NIH covalent and nonbonded energy terms were included in our refinement calculations. One hundred models were calculated and the ten sets of coordinates that best agreed with the experimental SAXS data−lowest $\chi^2$−were analyzed. The statistics for the conformers are reported in Supplementary Table 1.

## Nuclear magnetic resonance

NMR experiments were carried out using the UBE2D3 isoform as its assignment was available and kindly provided by Rachel Klevit. UBE2D3 shares over 88% sequence identity with the isoform UBE2D1 used for crystallization. $^{15}$N isotope enriched UBE2D3, bearing the mutation of the catalytic cysteine (C85) to serine, was prepared by growing the bacteria in M9 minimal medium using 1 g/L of $^{15}$N-ammonium chloride as sole source of nitrogen. UBE2D3 titration with the unlabeled RING domains of TRIM2 and TRIM3 (0 to 2 molar equivalents) were recorded as previously described[41] at 298 K at constant 200 μM concentration of labeled component on a Bruker AVANCE spectrometer operating at a proton nominal frequency of 800 MHz in the NMR buffer 25 mM Na-phosphate pH 7.0 and 150 mM NaCl. Data were acquired with Topspin (Bruker), processed with NMRPipe[64] and analyzed by CCPNMR[65]. Chemical shifts changes for the backbone amide proton and nitrogen nuclei ($\Delta\delta^{NH}$) and dissociation constants ($K_d$) for the RING/E2 complex (RING/E2 ↔ RING + E2) were calculated according to a procedure implemented in CCPNMR analysis[65]. Since both proteins were prepared in the same NMR buffer any change in the spectrum of the labeled E2 can be attributed directly to intermolecular interactions. Chemical shifts perturbations were mapped according to their value relative to the standard deviation of the $\Delta\delta^{NH}$ measurements. The data were plotted using GraphPad.

## Phylogenetic analysis

Using full-length human TRIM2 (Uniprot Q9C040) and TRIM3 (Uniprot O75382) protein sequences as queries, BLAST[66] was used to search for orthologous sequences at NCBI[67] and at Ensembl[68] (release 105). Muscle[69] and MAFFT[70] were used for sequence alignments. The alignments were inspected in Jalview[71]. The final alignment consisted of 75 vertebrate sequences spanning the seven extant phylogenetic classes. The coding DNA sequences corresponding to each of the protein entries were downloaded via in-house perl scripts and using Entrez Direct programs. A DNA sequence alignment was created from the amino acid alignment with RevTrans[72]. IQ-Tree 1.6.11[73] was used to infer the maximum-likelihood tree using 1000 ultrafast bootstrap replicates[74] for branch support. The best-fit codon model was SCHN05 + F + G4 as selected by Modelfinder[75]. Trees were inspected and prepared for figures with Dendroscope[76]. Secondary structure prediction was carried out using AlphaFold2[36].

## Reporting summary

Further information on research design is available in the Nature Portfolio Reporting Summary linked to this article.

## Data availability

The data that support this study are available from the corresponding author upon request. The structure reported has been deposited in the Protein Data Bank under accession code 7ZJ3. Other Protein Data Bank entries used in this study: 5FEY (TRIM32 RING); 5FER (TRIM25-RING/UBE2D1-Ub); 6YXE (TRIM69); 4TKP (TRIM5α/UBE2N) and 5EGG (UBE2D3). Source data are provided with this paper.

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

## Acknowledgements

The authors thank the Crick Structural Biology Science Technology Platforms for expert technical support, especially Phil Walker and Andy Purkiss; the Diamond Light Source at Oxford, UK for synchrotron access; the staff of the SWING beamline at SOLEIL at Gif-sur-Yvette CEDEX, France; Rachel Klevit, University of Washington, for sharing the NMR assignments of UBE2D3; Ian Taylor for help with SEC-MALLS experiments; Geoff Kelly, Alain Oregioni and Tom Frenkiel of the MRC Biomedical NMR Centre for access and advice. This work was supported by the Francis Crick Institute which receives its core funding from Cancer Research UK (CC2075, CC2000), the UK Medical Research Council (CC2075, CC2000), and the Wellcome Trust (CC2075, CC2000). For the purpose of Open Access, the author has applied a CC BY public copyright licence to any Author Accepted Manuscript version arising from this submission.

## Author contributions

D.E. carried out all biochemical, biophysical and crystallographic experiments, J.D.-F. carried out and analyzed cell and molecular biology experiments, A.G.-G. carried out the phylogenetic analysis, D.E. and K.R. designed the project, analyzed the data and wrote the paper with input from all authors.

## Funding

## Competing interests

The authors declare no competing interests.
