## [Peer Review File · Nature Communications]

REVIEWER COMMENTS

Reviewer #1 (Remarks to the Author):

Review of Esposito et al

Tunable Self-Association of the paralogous proteins Trim2 and Trim3 regulates their E3 ligase activity

Trim proteins are a large family of proteins sharing a conserved domain architecture that includes an N-terminal RING domain typically conferring activity. Dimerization of the RING domain is a key requirement for E3-Ubiquitin ligase activity in all members studied thus far. The authors present a structural and biochemical study of TRIM2 and TRIM3, members of the Class VII TRIM proteins examining the E3-ubiquitin ligase activity of each protein.

Ubiquitin discharge assays show a clear difference in activity between TRIM2 and TRIM3 despite a high level of sequence identity. The RING domain of TRIM3 in particular lacks activity, while there is partial activity in full length constructs. SEC MALLs analysis demonstrates that the RING domain of Trim3 is a monomer in solution providing a possible hypothesis why it is inactive. The RING dimerization or lack thereof is reflected in the assembly of the RBCC constructs for each protein studied in this article.

The authors present the structure of the TRIM2 RING dimer in complex with the E2 UBE2D1-Ub conjugate. This structure is similar to previously presented structure of TRIM RING domains in complex with an E2-Ub conjugate and probe residues required for activity. The structure has been well determined with no obvious errors presented.

I would question the precision of the unit cell dimensions presented in Table 1. There is no need to present 3 merging R factors (Rmerge, Rmeas, Rpim). CC1/2 cut-off for data selection is appropriate.

Disruption of RING dimerization abolishes activity while mutations impacting the E2 binding site or Ub binding site have limited effect. Mutations that mediate interactions between all three components at the interface located between the E2/Ub and RING abolish discharge activity.

To characterise the structure of the monomeric Trim3 domain the authors present SAXS data and use modelling to confirm Trim3 RING domain is folded as expected with a likely disordered N-terminal helix – a major contributor to the dimerization interface.

To complete this analysis the authors should include a plot showing the Radius of gyration of versus elution for the frames selected for analysis as a supplementary figure. This is particularly important for the Trim2 given the propensity for assembly as seen in SEC-MALLS. It would be useful for the modelling statistics in Table 2 to be separated or clearly associated with Trim3, at present they are located in the centre of the two columns.

To probe the reason for differences in activity between TRIM2 and TRIM3 the authors ask 2 key questions... is the RING domain of TRIM3 still able to interact with the E2 enzyme? And is the lack of dimerization the reason for lack of activity? Using NMR they demonstrate that TRIM3 is still able to interact with the E2 enzyme, with a tighter dissociation constant than the active TRIM2 suggesting that interaction with the E2 is not responsible for differences in activity.

To understand if dimerization is the reason the authors probe changes in the helical bundle that mediates dimerization of TRIM RING domains. Individual mutations to the have limited effect increasing activity, while corresponding mutations blocking dimerization of Trim2 reduce activity. Substituting the helical regions from TRIM2 onto the TRIM3 core returns results in activity equivalent to TRIM2 and fusion of two copies of the TRIM3 ring domain results in activity demonstrating that dimerization is the key feature missing for activity.

The authors then extend their claim to include heterodimerisation between TRIM2 and TRIM3. This claim is based upon two pieces of evidence 1) the T3R-T2R construct fusion showing activity. 2) Measurement of Co-IP and Ub activity in cells.

The T3R-T2R fusion results could be interpreted in ways other than hetero-association between the T2R and T3R.

Can the authors rule out that the observed activity is a result of a T2R dimer formed by self-association - rather than a heterodimer? It would be useful to see SEC-MALLS data for these constructs to assess whether there is additional assembly occurring in solution. Alternatively, if it is possible, disrupting the T2R activity within the T2R-T3R fusion would confirm that the T3R is activated in the fusion.

The Co-IP experiments demonstrate an interaction between TRIM2 and TRIM3 when overexpressed in cells in agreement with previous reports. Investigation of ubiquitination activity from these pulldowns show a small amount of activity from TRIM3 in the presence of TRIM2 and a reduction in TRIM3 activity in the presence of TRIM2. The reduction in activity is more convincing suggesting that Trim2 and Trim3 hetero-association might indeed regulate activity. If true, this is the first report I am aware of.

Overall, the study is well presented and provides new insight into what elements of TRIM proteins regulate the E3-Ubiquitin ligase activity. It is interesting to see that the RING domain of TRIM3 maintains the ability to recognise and bind the E2 enzyme with similar affinity to the active TRIM2 protein but the lack of dimerization prevents activity. The claims of hetero-association regulating activity are novel and very much warrant further investigation. This study sets the scene nicely for that work to occur.

Reviewer #2 (Remarks to the Author):

The manuscript by Esposito *et al.* used structural biology techniques (XRD and NMR, SAXS), biophysical techniques including SEC-MALLS, biochemical assays, mutagenesis, and bioinformatical analysis to study the structure and activity regulation of the tripartite motif (TRIM) family E3 ubiquitin ligases TRIM2 and TRIM3 and their interactions. Overall, this is a well-written manuscript with very solid results, which connects the interesting observation on the structure/activity to the evolution of these two E3s.

I only have a few comments the authors should address:

1. pg. 8 How the calculation of solvation free energy is not detailed, was it based on PISA analysis? The calculation result usually does not correlate the actual binding affinity and I would remove those numbers from the manuscript.
2. pg. 8, last paragraph talks about the I44-V70 patch, it would be good to include an graphic illustration (in Fig. 2?) to show the interface.
3. pg. 9, first paragraph discusses about the interaction between the residues, e.g. hydrogen bonds, please indicate network of interactions in the figure (Fig.2) using dashed bonds (PyMOL).
4. pg. 9, first paragraph last line. F81A mutation should cite Fig. 3C
5. pg. 10, first paragraph used the dimeric TRIM2 RING structure to fit the SAXS data and shows a χ^2 of 14. Since the authors know that TRIM2 has an equilibrium of dimer and monomer, can they calculate the population distribution of the two species and fit with the SAXS data again?

6. pg. 10 first paragraph, the SAXS for TRIM3 was fitted with an AlphaFold2 model and χ^2 is 4.4. I think it would be better to use the results of Xplor-NIH fitting (Fig. S3A) to replace the Fig 4B right panel and change the text accordingly. Label the N-terminus in the ab-initio envelop model.

7. pg. 13, interact of TRIM2 and TRIM3 in the cells. This is probably the weakest point of this manuscript: can the authors confirm which domains are responsible for the interaction of the two E3s using different truncated constructs? The authors did use in other parts a forced dimerization approach to fuse the TRIM2 and TRIM3 RING domains, but that is not physiological and does not clarify whether in the full length proteins the two protein dimerize through the RING domains or the coiled-coil regions. Further, the title of the article says "Tunable ...", by tunable it normally give the readers the sense that the binding affinity of the two can vary and be ****tuned****, however the manuscript does not describe such "tunability". I would recommend the authors to revise the title to reflect the content more precisely, or simply remove the "tunable" from the title

8. pg. 14, Figure 5D sequence alignment, please indicated the residues involved in Zinc-chelating in the alignment

9. pg. 14, alpha 1 of TRIM3 from fish to marsupials are predicted using JPRED4, I believe AlphaFold2 has higher reliability in prediction than JPRED4, so try it in AlphaFold2.

10. Indicate in Fig.5D that the secondary structure of TRIM2 is experimentally determined, while that for the TRIM3s are predicted (alphafold or JPRED4?)

11. pg. 22 line 4 from bottom "a Xplor-NIH" should be "an Xplor-NIH"

12. pg. 24 Data availability: other PDB codes used are not determined in this study and are in the public domain. Mentioned those in the corresponding figure legends instead of "Data availability" section.

13. Spelling: Space group (*P* 1 21 1) should be in italic

Reviewer #3 (Remarks to the Author):

The manuscript by Esposito and colleagues report the crystal structure of TRIM2 RING domain bound to E2-Ub and further structural analyses of TRIM2 and TRIM3 under the manuscript title of 'Tunable self-association of the paralogous protein TRIM2 and TRIM3 regulates their E3 ligase activity'.

The primary result is a crystal structure of TRIM2 RING bound to UBE2D1-Ub conjugate. Like other published structures of RING-E2-Ub, all critical interactions from RING (linchpin residue etc) and from E2 are identified in the current TRIM2 RING/ UBE2D1-Ub complex for the active E2-Ub closed conformation. The authors performed a neat work in the structural analyses of TRIM2 using several

methods such as SEC-MALS, SAXS, binding analysis with E2-Ub using CSPs and mutagenesis-ubiquitin assay, all findings corroborates each other and agrees well. The current structure data is highly similar to the previous findings from TRIM25 RING, TRIM23 RING and TRIM21 RING bound to UBE2x-Ub complex. The quality of table 1 and 2 is neat and well characterised.

The second part of this manuscript are the findings of TRIM3 full length possessing E3 ligase activity while the shorter constructs RING and RB are inactive. Not surprisingly, the chimeric versions of TRIM3 (T3R-T3R or T3R-T2R) shows similar catalytic activity as TRIM2 RING domain. Authors could not evidently explain the reason behind TRIM3 RING and RB being inactive while full length possess the catalytic activity.

My main criticism is the lack of strong evidence of physiological evidences of TRIM2-TRIM3 interactions in cellular context. Moreover the manuscript title is misleading by claiming to present the tunable self-association of the TRIM2 and TRIM3 regulating their E3 ligase activity. I am not convinced that the findings of co-immunoprecipitation presented provide sufficient evidences. Given the other TRIM RING/E2-Ub structures, the manuscript presenting only the structure and characterisation of TRIM2 bound to E2-Ub and TRIM3 alone does not provide conceptual advances for publication in Nature Communications Journal.

Specific comments:

1. The manuscript lacks the discussion/data of B-Box2 especially in context of TRIM3 RING and RB domain being inactive. Could the authors relate to the available structure of TRIM21 RB2 (pdb 5OLM) and dissect the mechanism? Notably, TRIM21 RB2 also possess auto-inhibitory effect.

2. To assess the quality of full length proteins of TRIM2 and TRIM3, could the authors kindly show the SDS-PAGE of the purified material?

3. For the NMR binding experiments authors used UbCH5c/UBE2D3 instead of the highly similar UBE2D1 (used in crystal structure) because of the available backbone assignments. Could the authors mention that these two E2s are highly similar in the methods?

4. minor thing: could the authors be consistent in using UBE2D1 instead of UbCH5a? like in Table 1.

Point-by-point response to Reviewers Comments

We thank the reviewers for their careful assessment of our manuscript and insightful comments. In response to the points raised we have carried out additional experiments, specifically focussing on the interaction between TRIM2 and TRIM3 in a cellular context. We now show that endogenous TRIM2 and TRIM3 do interact in brain-derived cell lines as shown by immune-precipitation and immunofluorescence, strongly supporting our notion that they may regulate each other's activity. Furthermore, we mapped the sites of interaction between TRIM2 and TRIM3 to their filamin domain and coiled coil region, respectively. To strengthen the observation that heterodimerization of the RINGS of TRIM2 and TRIM3 restores activity to TRIM3 we now provide SEC-MALLS experiments that show that no further self-association of the fused heterodimer occurs, clearly indicating that the observed activity is not from TRIM2 RING dimers formed in trans. Additionally, we introduce mutations into the heterodimer that affect activity of the individual RINGS, which further support our conclusions.

Below are our detailed responses to all points raised, highlighted in blue.

In addition to our response to reviewers' comments we have moved the section on the phylogenetic analysis of TRIM2/3 to fit with the description of additional experiments.

Reviewer #1

Trim proteins are a large family of proteins sharing a conserved domain architecture that includes an N-terminal RING domain typically conferring activity. Dimerization of the RING domain is a key requirement for E3-Ubiquitin ligase activity in all members studied thus far. The authors present a structural and biochemical study of TRIM2 and TRIM3, members of the Class VII TRIM proteins examining the E3-ubiquitin ligase activity of each protein.

Ubiquitin discharge assays show a clear difference in activity between TRIM2 and TRIM3 despite a high level of sequence identity. The RING domain of TRIM3 in particular lacks activity, while there is partial activity in full length constructs. SEC MALLS analysis demonstrates that the RING domain of Trim3 is a monomer in solution providing a possible hypothesis why it is inactive. The RING dimerization or lack thereof is reflected in the assembly of the RBCC constructs for each protein studied in this article.

The authors present the structure of the TRIM2 RING dimer in complex with the E2 UBE2D1-Ub conjugate. This structure is similar to previously presented structure of TRIM RING domains in complex with an E2-Ub conjugate and probe residues required for activity. The structure has been well determined with no obvious errors presented.

I would question the precision of the unit cell dimensions presented in Table 1. There is no need to present 3 merging R factors (Rmerge, Rmeas, Rpim). CC1/2 cut-off for data selection is appropriate.

Thank you for the comments on the structure quality. We wanted to be meticulous in reporting X-ray data statistics but fully agree that there is no need to report Rmerge. We have edited Table 1 accordingly and now only shown Rmeas and Rpim.

Disruption of RING dimerization abolishes activity while mutations impacting the E2 binding site or Ub binding site have limited effect. Mutations that mediate interactions between all three components at the interface located between the E2/Ub and RING abolish discharge activity.

To characterise the structure of the monomeric Trim3 domain the authors present SAXS data and use modelling to confirm Trim3 RING domain is folded as expected with a likely disordered N-terminal helix – a major contributor to the dimerization interface.

To complete this analysis the authors should include a plot showing the Radius of gyration of versus elution for the frames selected for analysis as a supplementary figure. This is particularly important for the Trim2 given the propensity for assembly as seen in SEC-MALLS. It would be useful for the modelling statistics in Table 2 to be separated or clearly associated with Trim3, at present they are located in the centre of the two columns.

Thank you. We have edited Figure S4 to include plots of the integrated scattering intensities and the radii of gyration along the SEC experiments for TRIM2 and TRIM3 RING. The plots also highlight the frames used for the analysis.

We have also modified the caption for the modelling statistics in Table 2 to clearly associate it with TRIM3 RING.

To probe the reason for differences in activity between TRIM2 and TRIM3 the authors ask 2 key questions... is the RING domain of TRIM3 still able to interact with the E2 enzyme? And is the lack of dimerization the reason for lack of activity? Using NMR they demonstrate that TRIM3 is still able to interact with the E2 enzyme, with a tighter dissociation constant than the active TRIM2 suggesting that interaction with the E2 is not responsible for differences in activity.

To understand if dimerization is the reason the authors probe changes in the helical bundle that mediates dimerization of TRIM RING domains. Individual mutations to the have limited effect increasing activity, while corresponding mutations blocking dimerization of Trim2 reduce activity. Substituting the helical regions from TRIM2 onto the TRIM3 core returns results in activity equivalent to TRIM2 and fusion of two copies of the TRIM3 ring domain results in activity demonstrating that dimerization is the key feature missing for activity.

The authors then extend their claim to include heterodimerisation between TRIM2 and TRIM3. This claim is based upon two pieces of evidence 1) the T3R-T2R construct fusion showing activity. 2) Measurement of Co-IP and Ub activity in cells.

The T3R-T2R fusion results could be interpreted in ways other than hetero-association between the T2R and T3R.

Can the authors rule out that the observed activity is a result of a T2R dimer formed by self-association - rather than a heterodimer? It would be useful to see SEC-MALLS data for these constructs to assess whether there is additional assembly occurring in solution. Alternatively, if it is possible, disrupting the T2R activity within the T2R-T3R fusion would confirm that the T3R is activated in the fusion.

We agree with the reviewer that this is a key experiment that was missing in our original submission. We have performed SEC-MALLS experiments of the RING fusion tandems and chimera construct and they are now reported in Figure S6. This experiment allow us to exclude T2R self-association in the fusion constructs as the MALLS data clearly show that both, T3R-T3R and T3R-T2R tandems, do not form dimers of the fusion proteins with a slight tendency for the T3R-T3R tandem to self-associate. The chimera construct T2 α s-T3Rcore has a SEC-MALLS profile similar to the T2RING construct at the same concentration.

To further strengthen this observation, we have performed lysine-discharged assays with T3R(F80A)-T2R and T3R-T2R(F81A) constructs and confirmed that activity in the fusion constructs relies on the ability of TRIM3 RING to bind the E2 and stabilise the ubiquitin in a closed conformation.

The Co-IP experiments demonstrate an interaction between TRIM2 and TRIM3 when overexpressed in cells in agreement with previous reports. Investigation of ubiquitination activity from these pulldowns show a small amount of activity from TRIM3 in the presence of TRIM2 and a reduction in TRIM3 activity in the presence of TRIM2. The reduction in activity is more convincing suggesting that Trim2 and Trim3 hetero-association might indeed regulate activity. If true, this is the first report I am aware of.

Overall, the study is well presented and provides new insight into what elements of TRIM proteins regulate the E3-Ubiquitin ligase activity. It is interesting to see that the RING domain of TRIM3 maintains the ability to recognise and bind the E2 enzyme with similar affinity to the active TRIM2 protein but the lack of dimerization prevents activity. The claims of hetero-association regulating activity are novel and very much warrant further investigation. This study sets the scene nicely for that work to occur.

Thank you for acknowledging the quality of our work and importance of our observation that TRIM2 and TRIM3 interact in cells and can regulate each other's activity.

Reviewer #2

The manuscript by Esposito _et al_ used structural biology techniques (XRD and NMR, SAXS), biophysical techniques including SEC-MALLS, biochemical assays, mutagenesis, and bioinformatical analysis to study the structure and activity regulation of the tripartite motif (TRIM) family E3 ubiquitin ligases TRIM2 and TRIM3 and their interactions. Overall, this is a well-written manuscript with very solid results, which connects the interesting observation on the structure/activity to the evolution of these two E3s.

Thank you for acknowledging the quality of our work.

I only have a few comments the authors should address:

1. pg. 8 How the calculation of solvation free energy is not detailed, was it based on PISA analysis? The calculation result usually does not correlate the actual binding affinity and I would remove those numbers from the manuscript.

The solvation free energies values were calculated using the PDBePISA server. We would prefer to keep these numbers in the manuscript because they relate to other studies on TRIM RING domains (Refs 29 and 32 of the manuscript).

2. pg. 8, last paragraph talks about the I44-V70 patch, it would be good to include an graphic illustration (in Fig. 2?) to show the interface.

Thank you for this suggestion. We agree and have provided the relevant illustrations for the E2-Ub (I44-V70 patch) and E2-RING interfaces in a new Supplementary figure (S3). We have also added a cartoon representation of the complex and highlighted the interfaces described in the manuscript.

3. pg. 9, first paragraph discusses about the interaction between the residues, e.g. hydrogen bonds, please indicate network of interactions in the figure (Fig.2) using dashed bonds (PyMOL).

We have updated Figure 3 accordingly, and now also show cartoons to better illustrate the relevant interfaces.

4. pg. 9, first paragraph last line. F81A mutation should cite Fig. 3C

Done

5. pg. 10, first paragraph used the dimeric TRIM2 RING structure to fit the SAXS data and shows a χ^2 of 14. Since the authors know that TRIM2 has an equilibrium of dimer and monomer, can they calculate the population distribution of the two species and fit with the SAXS data again?

We have initially used "Oligomer", in the ATSAS package, to obtain the TRIM2 RING monomer-dimer equilibrium populations from the fit of the SAXS data, using the monomeric and dimeric TRIM2 RING coordinates derived from our crystal structure as inputs. The procedure results in a χ^2 of 9.9 with very little contribution from the monomeric RING to the experimental scattering intensities.

In our opinion, the reason why we cannot correctly estimate the equilibrium species distribution by fitting the TRIM2 RING experimental X-ray scattering data, is the absence of sufficiently accurate input structural models for each component. Our TRIM2 RING construct has a few additional N- and C-terminal residues, which however are absent in the electronic density and very likely flexible, as also highlighted by the uptrend of the Kratky plot (Figure 4a). Furthermore, we believe it is conceivable that, as in the case of the TRIM3 RING, the monomeric TRIM2 RING possesses partially unfolded and dynamic N- and C-terminal α -helical segments which will fold into the four-helix bundle structure upon self-association.

6. pg. 10 first paragraph, the SAXS for TRIM3 was fitted with an AlphaFold2 model and χ^2 is 4.4. I think it would be better to use the results of Xplor-NIH fitting (Fig. S3A) to replace the Fig 4B right panel and change the text accordingly. Label the N-terminus in the ab-initio envelop model.

We have changed Figure 4 accordingly and now show the final Xplor-NIH ten-conformers bundle aligned to the ab-initio DAMMIF calculated SAXS envelope and changed the figure caption accordingly. The N- and C-terminus of the envelope/conformers have been labelled.

7. pg. 13, interact of TRIM2 and TRIM3 in the cells. This is probably the weakest point of this manuscript: can the authors confirm which domains are responsible for the interaction of the two E3s using different truncated constructs?

Thank you for this suggestion. In response to this comment and those from the other reviewers, we have investigated the interaction of TRIM2 and TRIM3 in cells in more detail. We now show that the two proteins interact and co-localise on an endogenous level by immunoprecipitation and immune fluorescence. We have also mapped the interaction between the two proteins and show that they form an asymmetric complex in which the filamin domain of TRIM2 binds the coiled coil region of TRIM3.

8. The authors did use in other parts a forced dimerization approach to fuse the TRIM2 and TRIM3 RING domains, but that is not physiological and does not clarify whether in the full length proteins the two protein dimerize through the RING domains or the coiled-coil regions. Further, the title of the article says "Tunable ...", by tunable it normally give the readers the sense that the binding affinity of the two can vary and be ****tuned****, however the manuscript does not describe such "tunability". I would recommend the authors to revise the title to reflect the content more precisely, or simply remove the "tunable" from the title

We agree that "tunable" is not the right word and might be misunderstood. We have therefore revised the title to "Divergent self-association properties of paralogous proteins TRIM2 and TRIM3 regulate their E3 ligase activity".

To further explore the interaction between TRIM2 and TRIM3 we have mapped the domains involved in complex formation. These involve the filamin domain of TRIM2 which binds the coiled-coil region of TRIM3. These data are now shown in Figure 6.

9. pg. 14, Figure 5D sequence alignment, please indicated the residues involved in Zinc-chelating in the alignment

The Figure has been updated to highlight the zinc-coordinating residues.

10. pg. 14, alpha 1 of TRIM3 from fish to marsupials are predicted using JPRED4, I believe AlphaFold2 has higher reliability in prediction than JPRED4, so try it in AlphaFold2.

We have now added the AlphaFold2 prediction. We would also like to highlight that this region of TRIM3 from fish to marsupials has the identical primary sequence as TRIM2 which adopts a helix as show in our crystal structure.

11. Indicate in Fig.5D that the secondary structure of TRIM2 is experimentally determined, while that for the TRIM3s are predicted (alphafold or JPRED4?)

This is now described in the Figure legend.

12. pg. 22 line 4 from bottom "a Xplor-NIH" should be "an Xplor-NIH"

Thank you. This has been changed.

13. pg. 24 Data availability: other PDB codes used are not determined in this study and are in the public domain. Mentioned those in the corresponding figure legends instead of "Data availability" section.

We have been asked by the editor to list these PDB codes in Data availability.

14. Spelling: Space group (*P* 1 21 1) should be in italic

Done.

Reviewer #3

The manuscript by Esposito and colleagues report the crystal structure of TRIM2 RING domain bound to E2-Ub and further structural analyses of TRIM2 and TRIM3 under the manuscript title of

'Tunable self-association of the paralogous protein TRIM2 and TRIM3 regulates their E3 ligase activity'.

The primary result is a crystal structure of TRIM2 RING bound to UBE2D1-Ub conjugate. Like other published structures of RING-E2-Ub, all critical interactions from RING (linchpin residue etc) and from E2 are identified in the current TRIM2 RING/ UBE2D1-Ub complex for the active E2-Ub closed conformation. The authors performed a neat work in the structural analyses of TRIM2 using several methods such as SEC-MALS, SAXS, binding analysis with E2-Ub using CSPs and mutagenesis-ubiquitin assay, all findings corroborates each other and agrees well. The current structure data is highly similar to the previous findings from TRIM25 RING, TRIM23 RING and TRIM21 RING bound to UBE2x-Ub complex. The quality of table 1 and 2 is neat and well characterised.

The second part of this manuscript are the findings of TRIM3 full length possessing E3 ligase activity while the shorter constructs RING and RB are inactive. Not surprisingly, the chimeric versions of TRIM3 (T3R-T3R or T3R-T2R) shows similar catalytic activity as TRIM2 RING domain. Authors could not evidently explain the reason behind TRIM3 RING and RB being inactive while full length possess the catalytic activity.

My main criticism is the lack of strong evidence of physiological evidences of TRIM2-TRIM3 interactions in cellular context.

We agree that the data of TRIM2 and TRIM3 interaction in a cellular context were solely based on overexpression studies and hence not as strong as they could be. Therefore, to further support our findings we carried out additional experiments to evaluate a possible TRIM2-TRIM3 interaction on an endogenous level.

Firstly, we were able to show that TRIM2 and TRIM3 are well expressed in a set of brain-derived cell lines (Fig. 5d). We then used these lines to demonstrate that endogenous TRIM2 and TRIM3 co-immunoprecipitate and co-localise (Fig. 5e).

Interestingly, these proteins localised to protrusions at the cell periphery, possibly suggestive of a link to known TRIM2 function in regulating axonal outgrowth via neurofilament turnover (Balastik et al, *PNAS*, 2008; Ylikallio et al, *Hum Mol Gen*, 2013).

Furthermore, we have now mapped the regions that mediate the interaction between the two proteins.

Moreover the manuscript title is misleading by claiming to present the tunable self-association of the TRIM2 and TRIM3 regulating their E3 ligase activity. I am not convinced that the findings of co-immunoprecipitation presented provide sufficient evidences.

Apologies. We have changed the title to:

“Divergent self-association properties of paralogous proteins TRIM2 and TRIM3 regulate their E3 ligase activity”

Given the other TRIM RING/E2-Ub structures, the manuscript presenting only the structure and characterisation of TRIM2 bound to E2-Ub and TRIM3 alone does not provide conceptual advances for publication in Nature Communications Journal.

Specific comments:

1. The manuscript lacks the discussion/data of B-Box2 especially in context of TRIM3 RING and RB domain being inactive. Could the authors relate to the available structure of TRIM21 RB2 (pdb 5OLM) and dissect the mechanism? Notably, TRIM21 RB2 also possess auto-inhibitory effect.

Thank you for prompting us to further clarify the role of the Bbox2 in TRIM2/3 activity. There is currently no evidence that the Bbox2 plays, *in vitro*, a role in regulating the activity of either TRIM2 or TRIM3. It does not influence self-association, nor does it affect catalytic activity.

The reviewer is correct in pointing out that the Bbox2 in TRIM21 has an auto-inhibitory effect by occupying the E2 binding site of the RING domain. Removal of the inhibition in TRIM21 is achieved by RING phosphorylation, which displaces the Bbox2 and allows the binding of the E2~Ub conjugate. This mechanism is supported by the observation that the TRIM21 RB2 construct has a much-reduced rate in auto-ubiquitination and lysine discharge experiments compared to the isolated RING which should have full activity. Conversely, in our auto-ubiquitination and ubiquitin lysine discharge experiments, we do not observe a difference in the behaviour and activity of the RB2 construct compared to the isolated RING of TRIM3, strongly indicating that the presence of the B-box does not affect activity. Furthermore, our SEC-MALLS data do not provide evidence for a role of the B-box in higher order association.

We have now added a sentence to the Discussion to highlight this observation.

2. To assess the quality of full length proteins of TRIM2 and TRIM3, could the authors kindly show the SDS-PAGE of the purified material?

We have now show the Coomassie-stained gel of an auto-ubiquitination assay with the full length constructs in Figure S1c that highlights at time 0 min the purity of TRIM2 and TRIM3 proteins. Furthermore, full length recombinant TRIM2 and TRIM3 proteins were also run on SDS-PAGE gels at varying concentrations and their quality assessed by western blotting. This data has been added as Supplementary Figure S6a.

3. For the NMR binding experiments authors used UbcH5c/UBE2D3 instead of the highly similar UBE2D1 (used in crystal structure) because of the available backbone assignments. Could the authors mention that these two E2s are highly similar in the methods?

Thank you for pointing out this omission. We have added a sentence in the NMR methods section highlighting the similarity between the two E2 isoforms.

4. Minor thing: could the authors be consistent in using UBE2D1 instead of UbCH5a? like in Table 1. Thank you for noticing this. We have corrected the heading of Table 1.

A key consideration during the additional experiments presented in our rebuttal was the potential for TRIM2 and TRIM3 antibodies to cross-react with both TRIM2 and TRIM3, due to their high sequence homology. The TRIM3 antibody from Abcam (ab111840) was previously shown to act specifically against TRIM3.

Rebuttal Table 1 details the TRIM2 antibodies which we have tried. Supplementary Figure S6A shows that Protein Tech anti-TRIM2 (#16819925) reacts highly specifically with TRIM2, with very minimal cross-reactivity with TRIM3 that is only visible when the blot is excessively overexposed.

Moreover, siRNA knockdown followed by immunofluorescence (Rebuttal Figure 1A) and Western blotting (Rebuttal Figure 1B) demonstrated that TRIM2 and TRIM3 cellular localisation patterns and western blot bands were specific for each respective target protein.

These results gave us confidence in the accuracy of the findings of our co-immunoprecipitation and co-localisation experiments using these antibodies.

Rebuttal Table 1

Table detailing the TRIM2 antibodies tested, their advertised applications, and the immunogen homology with TRIM3.

Order info	Immunogen (and sequence identity with TRIM3)	Applications
Merck / Atlas Antibodies #HPA035854	KASLQVQLDAVNKRLPEIDSALQFISEIIHQLTNQKASIVDDI HSTFDELQKTLNVRKSVLLMELEVNYGLKHK (39% identity T3))	IF, IHC
Merck #SAB2501997	STFDELQKTLNVRKS (41% identity T3)	ELISA, WB
ProteinTech #16819925	TRIM2 fusion protein Ag14637 (unknown % identity)	IHC, IF, ELISA, WB

Rebuttal Figure 1

Validation of antibodies using siRNA in LN229 cells. A: Immunofluorescence against TRIM2 (magenta) and TRIM3 (green) in wild-type (WT) cells or those transfected with siRNAs against TRIM2 or TRIM3 (Merck, EHU073041 or EHU088111, respectively), B: Western blotting analysis of TRIM2, TRIM3, and GAPDH in WT or cells transfected with siRNAs against TRIM2 or TRIM3.

REVIEWERS' COMMENTS

Reviewer #1 (Remarks to the Author):

The Authors have addressed all my comments and concerns clearly and effectively. I would recommend the article for publication.

I have one minor comment for the authors regarding the revision:

In the supplemental figure 6 the masses are marked and "Dim" for the Trim2-3 fusions... should these be labeled and monomer? It would perhaps aid clarity for the reader to report masses of the fusion constructs in the legend.

Reviewer #2 (Remarks to the Author):

The authors have run additional experiments to confirm the interaction of TRIM2 and TRIM3 in a cellular context and mapped the interaction domains.

The authors have satisfactorily addressed all my questions in the revision. The work is solid and should be published.

Reviewer #3 (Remarks to the Author):

In this revised manuscript, the authors report the crystal structure of TRIM2 RING bound to UBE2D1-Ub conjugate and evidence of self-associating TRIM2 and TRIM3 for regulating their E3

ligase activity. In the revised version the authors show evidence of TRIM2-TRIM3 interaction on an endogenous level. Further, mapped the regions that mediate the interaction between these two

proteins using co-immunoprecipitation. It is quite fascinating to see how B-box2 possess different roles in different TRIM proteins. TRIM proteins are an important enzyme in health and disease, and

the structures, together with a wealth of biophysical and biochemical data, provide important and comprehensive insight into its function. The authors have addressed my comments appropriately.

Minor thing:

In the rebuttal letter, answering reviewer 3, I honestly believe the figure callouts are incorrect. Hope I am looking at the right thing, please check carefully.

1. Page 6. TRIM2-TRIM3 interaction on endogenous level. Instead of Figure 5d and 5e, it should be Fig 6.
2. Page 7. Showing the purity of proteins. I believe the authors want to refer Figure S1b and Figure S7c instead of Figure S1c and Figure S6a.

Given this, I request authors kindly revalidate the figure callouts in the main manuscript.

We thank all reviewers for considering our revised manuscript and their supportive responses.

Reviewer #1 (Remarks to the Author):

The Authors have addressed all my comments and concerns clearly and effectively. I would recommend the article for publication.

I have one minor comment for the authors regarding the revision:

In the supplemental figure 6 the masses are marked and "Dim" for the Trim2-3 fusions... should these be labeled and monomer? It would perhaps aide clarity for the reader to report masses of the fusion constructs in the legend.

We agree that the labelling wasn't sufficiently clear. We have now added the molecular masses to the figure legends and have changed the label in the figures to "fusion construct".

Reviewer #2 (Remarks to the Author):

The authors have run additional experiments to confirm the interaction of TRIM2 and TRIM3 in a cellular context and mapped the interaction domains.

The authors have satisfactorily addressed all my questions in the revision. The work is solid and should be published.

Reviewer #3 (Remarks to the Author):

In this revised manuscript, the authors report the crystal structure of TRIM2 RING bound to UBE2D1-Ub conjugate and evidence of self-associating TRIM2 and TRIM3 for regulating their E3 ligase activity. In the revised version the authors show evidence of TRIM2-TRIM3 interaction on an endogenous level. Further, mapped the regions that mediate the interaction between these two proteins using co-immunoprecipitation. It is quite fascinating to see how B-box2 possess different roles in different TRIM proteins. TRIM proteins are an important enzyme in health and disease, and the structures, together with a wealth of biophysical and biochemical data, provide important and comprehensive insight into its function. The authors have addressed my comments appropriately.

Minor thing:

In the rebuttal letter, answering reviewer 3, I honestly believe the figure callouts are incorrect. Hope I am looking at the right thing, please check carefully.

1. Page 6. TRIM2-TRIM3 interation on endogenous level. Instead of Figure 5d and 5e, it should be Fig 6.

2. Page 7. Showing the purity of proteins. I believe the authors want to refer Figure S1b and Figure S7c instead of Figure S1c and Figure S6a.

Given this, I request authors kindly revalidate the figure callouts in the main manuscript.

We apologise for this mistake. We have carefully checked all figure callouts in the manuscript to ensure that they are correct.